# GDF15 is required for cold-induced thermogenesis and contributes to improved systemic metabolic health following loss of OPA1 in brown adipocytes

**Jayashree Jena[1†], Luis Miguel García-Peña[1†], Eric T Weatherford[1], Alex Marti[1], Sarah H Bjorkman[1], Kevin Kato[1], Jivan Koneru[1], Jason H Chen[1], Randy J Seeley[2], E Dale Abel[1‡], Renata O Pereira[1]\***

[1]Fraternal Order of Eagles Diabetes Research Center and Division of Endocrinology and Metabolism, Roy J. and Lucille A. Carver College of Medicine, University of Iowa, Iowa City, United States; [2]Department of Internal Medicine, University of Michigan, Ann Arbor, United States

**\*For correspondence:**
renata-pereira@uiowa.edu

[†]These authors contributed equally to this work

**Present address:** [‡]Department of Medicine, David Geffen School of Medicine, University of California, Los Angeles, Los Angeles, United States

## Abstract

We previously reported that mice lacking the protein optic atrophy 1 (OPA1 BKO) in brown adipose tissue (BAT) display induction of the activating transcription factor 4 (ATF4), which promotes fibroblast growth factor 21 (FGF21) secretion as a batokine. FGF21 increases metabolic rates under baseline conditions but is dispensable for the resistance to diet-induced obesity (DIO) reported in OPA1 BKO mice (Pereira et al., 2021). To determine alternative mediators of this phenotype, we performed transcriptome analysis, which revealed increased levels of growth differentiation factor 15 (GDF15), along with increased protein kinase R (PKR)-like endoplasmic reticulum kinase (PERK) levels in BAT. To investigate whether ATF4 induction was mediated by PERK and evaluate the contribution of GDF15 to the resistance to DIO, we selectively deleted PERK or GDF15 in OPA1 BKO mice. Mice with reduced OPA1 and PERK levels in BAT had preserved ISR activation. Importantly, simultaneous deletion of OPA1 and GDF15 partially reversed the resistance to DIO and abrogated the improvements in glucose tolerance. Furthermore, GDF15 was required to improve cold-induced thermogenesis in OPA1 BKO mice. Taken together, our data indicate that PERK is dispensable to induce the ISR, but GDF15 contributes to the resistance to DIO, and is required for glucose homeostasis and thermoregulation in OPA1 BKO mice by increasing energy expenditure.

## Editor's evaluation

This article's findings are timely as the hormone GDF15 is being widely studied as a potential obesity and diabetes target. The results will add to this growing literature.

## Introduction

The integrated stress response (ISR) is a pro-survival signaling pathway present in eukaryotic cells, which is activated in response to a range of physiological and pathological stressors. Such stresses commonly include cell-extrinsic factors such as hypoxia, amino acid deprivation, glucose deprivation, and viral infection. However, cell-intrinsic stresses such as endoplasmic reticulum (ER) stress (*Pakos-Zebrucka et al., 2016*), and mitochondrial stress can also activate the ISR (*Bao et al., 2016*; *Quirós*

*et al., 2017*; *Pereira et al., 2021*). Induction of the ISR and its main effector activating transcription factor 4 (ATF4) in response to various stress conditions frequently correlates with increased levels of the endocrine factors fibroblast growth factor 21 (FGF21) and growth differentiation factor 15 (GDF15) in tissues, such as liver (*Kang et al., 2021*) and skeletal muscle (*Ost et al., 2020*). However, the upstream mechanisms mediating ISR induction in response to mitochondrial stress are unclear. Furthermore, the respective roles of FGF21 and GDF15 in the context of mitochondrial dysfunction are incompletely understood.

We recently reported that mitochondrial stress caused by deletion of the mitochondrial fusion protein optic atrophy 1 (OPA1 BKO) activated the ISR and induced its main effector ATF4 in brown adipose tissue (BAT). ATF4-mediated FGF21 induction was required to promote browning of inguinal white adipose tissue (iWAT) and improve thermoregulation in OPA1 BKO mice at baseline conditions. Although activation of the ISR in BAT led to resistance to diet-induced obesity (DIO) in an ATF4-dependent manner, this phenomenon was independent of FGF21 (*Pereira et al., 2021*). In the present study, we sought to investigate the upstream mechanisms mediating ISR activation in BAT in response to OPA1 deletion and the molecular mechanisms downstream of ATF4 mediating resistance to DIO.

Transcriptome analysis showed that the unfolded protein response (UPR), and specifically the protein kinase R (PKR)-like ER kinase (PERK), was induced in OPA1-deficient BAT. Given that PERK is shared by the ISR and the UPR (*Pakos-Zebrucka et al., 2016*), here, we tested the hypothesis that PERK is required for ATF4 induction in BAT. We also observed that in addition to FGF21, GDF15 was highly induced in BAT upon OPA1 deletion. Because GDF15 has been shown to regulate energy homeostasis in rodents (*Tsai et al., 2018a*; *Chung et al., 2017b*), here we tested the hypothesis that GDF15 is downstream of ATF4 and mediates the resistance to DIO in mice lacking OPA1 in thermogenic adipocytes. Our data reveal that PERK is dispensable for ISR and ATF4 induction in BAT in response to OPA1 deletion. Notably, GDF15 partially mediates the resistance to DIO, but is necessary for the improvement in glucose homeostasis and hepatic steatosis following high-fat feeding in OPA1 BKO mice. Furthermore, mice lacking GDF15 were unable to properly thermoregulate during cold exposure. Mechanistically, GDF15 deficiency abrogated browning of iWAT under obesogenic conditions and prevented the induction of calcium cycling in muscle, which likely contributes to increased weight gain and impaired thermogenesis in OPA1 BKO mice. These data underscore the complex regulation of systemic metabolism by various BAT-derived endocrine regulators that are released in response to mitochondrial stress and highlight novel roles for BAT-derived GDF15 in the regulation of energy homeostasis.

## Results
### OPA1 BKO mice have higher resting metabolic rates and are completely resistant to DIO

We have previously shown that OPA1 BKO mice have reduced body mass, increased resting metabolic rates, and are resistant to DIO (*Pereira and McFarlane, 2019*). To further characterize their metabolic phenotype, we analyzed indirect calorimetry data collected in 12-week-old male mice while housed under thermoneutral conditions and after 12 wk of high-fat feeding. Under isocaloric conditions and after being housed at thermoneutral conditions (30°C) for 7 d, 12-week-old OPA1 BKO mice had reduced body mass (*Figure 1A*), with no detectable changes in food intake (*Figure 1B* and *Figure 1—figure supplement 1A*) or locomotor activity (*Figure 1C* and *Figure 1—figure supplement 1B*). Oxygen consumption (*Figure 1D*) and VCO$_2$ production (*Figure 1E*) were unchanged between genotypes, regardless of time of day. Interestingly, respiratory exchange ratio was significantly higher in KO mice, indicating an increased relative preference for carbohydrates as energy substrate (*Figure 1F* and *Figure 1—figure supplement 1C*). Although the averaged oxygen consumption rates (OCRs) (*Figure 1D*) and energy expenditure (*Figure 1G* and *Figure 1—figure supplement 1D*) values for the dark and light cycles were unchanged between genotypes, ANCOVA (*Figure 1H*) detected a significant group effect for the relationship between body mass and energy expenditure, indicating that the reduced body mass in KO mice is a result of higher metabolic rates. In response to high-fat feeding, weight gain is completely prevented in KO mice as shown in the body weight curve over time (*Figure 1I*). A subset of OPA1 BKO mice exemplifying this drastic abrogation of DIO (*Figure 1J*) was

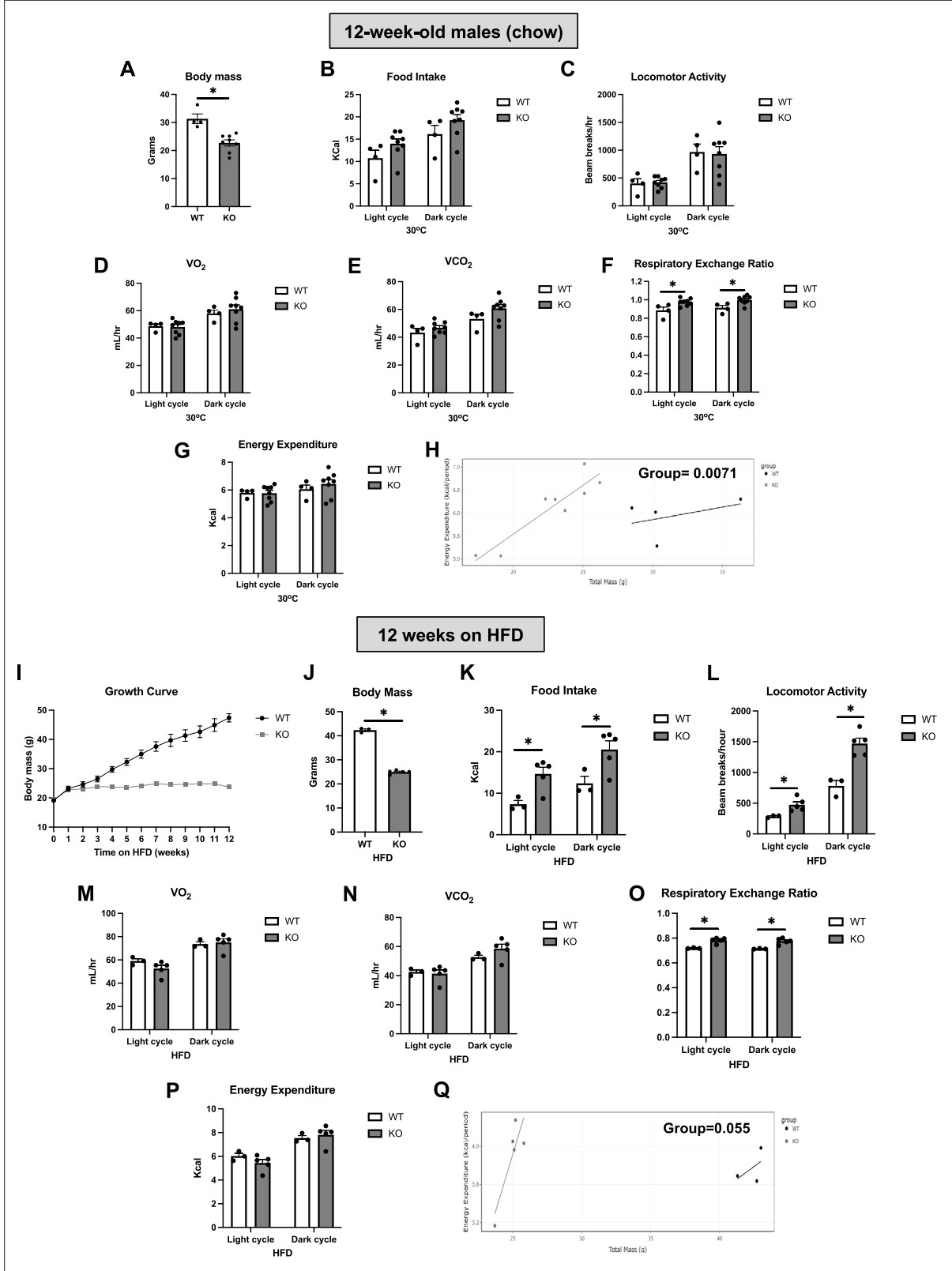

**Figure 1.** OPA1 BKO mice have higher resting metabolic rates and are completely resistant to diet-induced obesity (DIO). (**A–D**) Data collected on 12-week-old male OPA1 BKO mice. (**A**) Body mass. (**B–G**) Indirect calorimetry data represented as the average for the light and dark cycles during the last 48 hr of data recording. (**B**) Food intake. (**C**) Locomotor activity. (**D**) Oxygen consumption (VO$_2$). (**E**) Carbon dioxide production (VCO$_2$). (**F**) Respiratory exchange ratio. (**G**) Energy expenditure. (**H**) ANCOVA of energy expenditure as a function of body mass. (**I–Q**) Data collected on 18-week-

*Figure 1 continued on next page*

*Figure 1 continued*

old male OPA1 BKO mice fed high-fat diet (60% calories from fat) for 12 wk (diet started at 6 wk of age). (**I**) Body weight curve. (**J**) Final body weight (cohort used for indirect calorimetry data collection). (**K–Q**) Indirect calorimetry data represented as the average for the light and dark cycles during the last 48 hr of data recording. (**K**) Food intake. (**L**) Locomotor activity. (**M**) Oxygen consumption ($VO_2$). (**N**) Carbon dioxide production ($VCO_2$). (**O**) Respiratory exchange ratio. (**P**) Energy expenditure. (**Q**) ANCOVA of energy expenditure as a function of body mass. Data are expressed as means ± SEM. Significant differences were determined by Student's *t*-test using a significance level of p<0.05. *Significantly different vs. wild-type (WT) mice.

The online version of this article includes the following figure supplement(s) for figure 1:

**Figure supplement 1.** Food consumption, locomotor activity, and energy expenditure in OPA1 BAT knockout (KO) mice under baseline conditions and following high-fat feeding.

utilized for analysis of indirect calorimetry. Different than in our previously published study (*Pereira and McFarlane, 2019*), here we analyzed only mice fed high-fat diet (HFD), instead of simultaneously analyzing both HFD and control groups. Moreover, rather than representing the overall averaged data, we plotted the averaged data for the last 48 hr of data collection for each cycle of the day (light and dark). Surprisingly, although our previous analysis was unable to detect these changes, here we show that OPA1 BKO mice fed HFD for 12 wk had increased food intake (*Figure 1K* and *Figure 1—figure supplement 1E and F*) and locomotor activity (*Figure 1L* and *Figure 1—figure supplement 1G*) relative to their wild-type (WT) counterparts. Oxygen consumption (*Figure 1M*) and $VCO_2$ production (*Figure 1N*) were unchanged between genotypes, and respiratory exchange ratio was elevated in KO mice fed HFD (*Figure 1O* and *Figure 1—figure supplement 1H*). Although the absolute energy expenditure values were unchanged between genotypes (*Figure 1P* and *Figure 1—figure supplement 1I*), ANCOVA revealed a leftward shift in the relationship between body weight and energy expenditure in OPA1 BKO mice with a nearly significant group effect (p=0.055) (*Figure 1Q*). Indeed, this was also the case when the relationship between body weight and oxygen consumption was determined by ANCOVA, which revealed a statistically significant group effect (*Figure 1—figure supplement 1J*). Thus, relative to total body mass, KO mice have higher energy expenditure and OCRs compared to WT animals. Together, this data confirms the hypothesis we put forth in our previous study (*Pereira and McFarlane, 2019*) that the increase in resting metabolic rates likely contributes to the resistance to DIO in OPA1 BKO mice.

## Transcriptome analysis reveals induction of the UPR and GDF15 in mice lacking OPA1 in BAT

To gain insight into the molecular mechanisms underlying the favorable metabolic profile of OPA1 BKO mice, we performed bulk RNA sequencing (RNASeq) in BAT collected from 7-week-old male and female mice lacking OPA1 in thermogenic adipocytes (OPA1 BKO) and their respective WT littermate controls fed regular chow under ambient temperature conditions. Specifically, we were interested in identifying the mechanistic link between OPA1 deletion and induction of the ISR effector ATF4, and the molecular mechanism downstream of ATF4 that mediated resistance to DIO in OPA1 KO mice. Ingenuity Pathway Analysis revealed that the ER stress pathway and the UPR were amongst the top 3 induced canonical pathways in OPA1-deficient BAT (*Figure 2A*). *Fgf21*, which we previously demonstrated was required for activation of browning of iWAT and to regulate changes in core body temperature at baseline conditions, but not for the resistance to DIO in OPA1 BKO mice (*Pereira et al., 2021*), and *Gdf15* were among the top 25 upregulated genes in BAT in response to *Opa1* deletion (*Figure 2B*). The ER kinase PERK was significantly enriched in OPA1 BKO mice (see the GEO database under the accession number GSE218907), as shown in the bar graph comparing the averaged transcripts per million (TPM) for WT and KO mice (*Figure 2C*). Furthermore, *Gdf15* mRNA induction in OPA1 BKO mice was confirmed by qPCR (*Figure 2D*). Interestingly, mice lacking both ATF4 and OPA1 (OPA1/ATF4 DKO mice) in BAT (*Figure 2D*) had significantly lower levels of *Gdf15* mRNA expression relative to OPA1 BKO mice. This data suggests that ATF4 at least partially regulates *Gdf15* induction in OPA1-deficient BAT. Importantly, GDF15 serum levels were also induced in OPA1 BKO mice but were significantly attenuated in mice lacking both OPA1 and ATF4 in BAT (OPA1/ATF4) (*Figure 2E*). Conversely, GDF15 serum levels remained elevated in mice concomitantly lacking OPA1 and FGF21 in BAT (*Figure 2F*). These are important control groups as mice lacking both OPA1 and ATF4 in BAT are no longer resistant to DIO, while mice lacking OPA1 and FGF21 in BAT remain lean when fed HFD (*Pereira et al., 2021*). It is noteworthy that the degree of GDF15 induction observed in

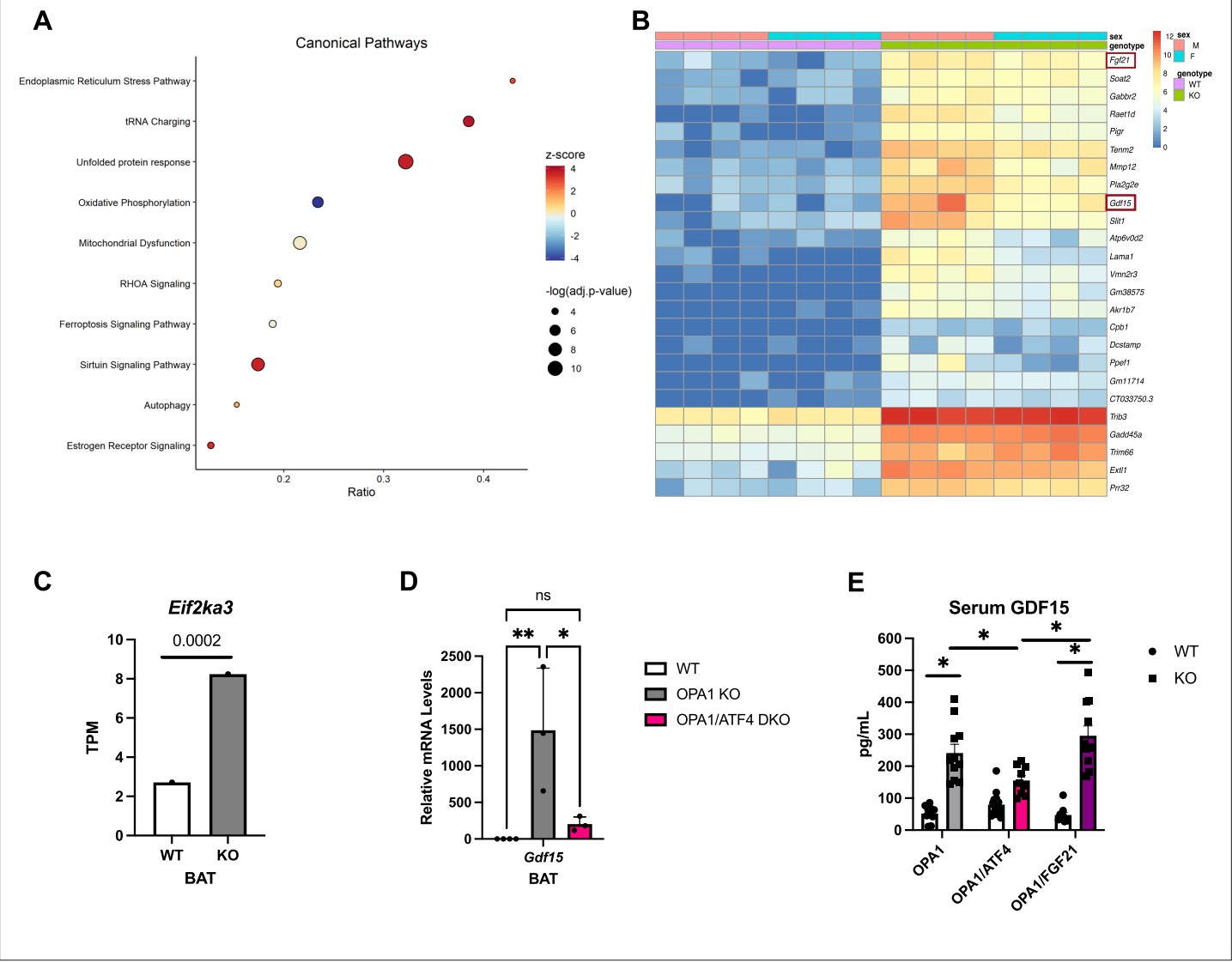

**Figure 2.** Transcriptome analysis reveals induction of the unfolded protein response (UPR) and GDF15 in mice lacking OPA1 in brown adipose tissue (BAT). (**A–C**) Data collected from BAT of 7-week-old male and female OPA1 BKO mice. (**A**) Bubble plot showing the top 10 canonical pathways from the Ingenuity Pathway Analysis (IPA) database containing genes with a significant overlap (adjusted p-value≤0.05) to those differentially expressed in OPA1 BKO mice. Size of the bubble indicates the -log of the adjusted p-value (Benjamini–Hochberg) from the pathway analysis. Plotted on the x-axis is the ratio of differentially expressed genes relative to the number of genes in the pathway. The bubble color indicates the z-score, which indicates the predicted activation (positive) or inhibition (negative) of the pathway based on the directionality of the gene changes in OPA1 BKO relative to wild-type (WT) mice. (**B**) Heatmap of the top 25 differentially expressed genes in OPA1 BKO mice. (**C**) Bar graph showing the Log2 transcript per million (TPM) for the *Eif2ka3* gene (encoding PERK). (**D**) Relative mRNA expression of *Gdf15* in BAT of OPA1 BKO and in OPA1/ATF4 BAT DKO mice normalized to tata box protein (*Tbp*) expression. (**E**) GDF15 serum levels in OPA1 BKO, OPA1/ATF4 BAT DKO, and OPA1/FGF21 BAT DKO mice. Data are expressed as means ± SEM. Significant differences were determined by Student's t-test using a significance level of p<0.05. * p <0.05; ** p <0.01. Significantly different vs. WT mice or from OPA1 KO.

OPA1 BKO mice is similar to that observed after cold exposure or prolonged high-fat feeding and is not further increased by these stressors (*Figure 1—figure supplement 1K*). Together, these data led us to hypothesize that PERK could be the ISR kinase regulating ATF4 induction, and that GDF15 is a downstream target of ATF4 mediating the resistance to DIO in OPA1 BKO mice.

## PERK is dispensable for the ISR activation in OPA1 BKO mice

Because the ER kinase PERK is shared by the UPR and the ISR, and is transcriptionally induced in OPA1 BKO mice (*Pakos-Zebrucka et al., 2016*), we hypothesized that activation of the PERK arm

of the UPR is required to induce the ISR and ATF4 in OPA1 BKO mice. To test this hypothesis, we generated mice concomitantly lacking both OPA1 and PERK in thermogenic adipocytes (OPA1/PERK BAT DKO) (*Figure 3A*). When we downregulated PERK expression in BAT of OPA1 BKO mice, phosphorylation of the translation initiation factor eIF2α, which is required for ATF4 translation, remained significantly elevated (*Figure 3B*). Accordingly, expression of the ISR genes *Atf4*, *Ddit3*, *Fgf21*, and *Gdf15* was induced in BAT (*Figure 3C*), which correlated with a significant increase in GDF15 serum levels (*Figure 3D*). As observed in OPA1 BKO mice (*Pereira et al., 2021*), expression of thermogenic genes was attenuated in BAT under baseline conditions (room temperature and regular chow feeding) (*Figure 3E*), and UCP1 protein levels were significantly reduced (*Figure 3F*). State 3 pyruvate/malate-supported mitochondrial respirations were significantly impaired in mitochondria isolated from DKO BAT (*Figure 3G*). Next, we investigated whether PERK deletion in the OPA1 BKO background would affect browning of iWAT. Expression of thermogenic genes (*Figure 3H*) was induced in the iWAT of DKO mice. Consistently, uncoupling protein 1 (UCP1) levels (*Figure 3I*) and tyrosine hydroxylase (TH) levels (*Figure 3J*) were also elevated in iWAT of OPA1/PERK BAT DKO mice, indicating increased compensatory browning. Metabolically, OPA1/PERK BAT DKO mice had reduced body mass (*Figure 3K*) and total fat mass (*Figure 3L*), with unchanged total lean mass (*Figure 3M*) at 20 wk of age under baseline conditions. These changes were the same as those observed in OPA1 BKO mice (*Pereira and McFarlane, 2019*). Together, these data suggest that PERK is dispensable for ATF4 induction in BAT or for the changes in energy metabolism, resulting in reduced body mass in OPA1 BKO mice under baseline conditions. Therefore, alternative ISR kinases are likely required to activate the ISR and its main effector ATF4 in BAT in response to OPA1 deletion.

## OPA1/PERK BAT DKO mice are resistant to DIO and insulin resistance

Although attenuation of PERK expression did not affect activation of the ISR in BAT, or the baseline changes in body composition observed in OPA1 BKO mice, we sought to determine whether simultaneous deletion of PERK and OPA1 in BAT would attenuate the resistance to DIO observed in OPA1 BKO mice. Therefore, we fed OPA1/PERK BAT DKO mice HFD for 12 wk. Like OPA1 BKO mice (*Pereira et al., 2021*), DKO mice were completely resistant to DIO, as demonstrated by overlapping weight curves between DKO and OPA1 BKO mice (*Figure 4A*). Relative to HFD controls, final body mass (*Figure 4B*) and percent fat mass were reduced (*Figure 4C*), and percent lean mass increased in DKO mice (*Figure 4D*), indicating reduced body mass at the expense of reduced fat mass. In contrast to OPA1 BKO mice, food intake was statistically unchanged between DKO mice and their WT littermate controls, although it trended higher (*Figure 4E* and *Figure 4—figure supplement 1A and B*). Locomotor activity was increased in DKO mice, as was the case in OPA1 BKO mice (*Figure 4F* and *Figure 4—figure supplement 1C and D*). Oxygen consumption (*Figure 4G*) and VCO$_2$ (*Figure 4H*) were unchanged between genotypes, regardless of the time of the day. Interestingly, in contrast to OPA1 BKO mice fed HFD, respiratory exchange ratio was significantly lower in DKO mice during the light cycle, indicating a relative increased preference for fatty acids as energy substrate (*Figure 4I* and *Figure 4—figure supplement 1E*). The averaged values for energy expenditure were unchanged between genotypes (*Figure 4J* and *Figure 4—figure supplement 1F*). However, ANCOVA revealed a leftward shift in the relationship of energy expenditure as a function of body mass in the OPA1/PERK BAT DKO mice, which was significantly different for the group effect (*Figure 4K*). These data indicate that increased resting metabolic rates in OPA1/PERK BAT DKO mice, as in OPA1 BKO mice, likely contribute to the reduced body weight observed in these mice. OPA1 BKO mice had significantly improved glucose homeostasis and insulin tolerance after 12 wk of HFD (*Pereira et al., 2021*). Likewise, glucose (*Figure 4L and M*) and insulin intolerance (*Figure 4N and O*) were ameliorated in high-fat-fed OPA1/PERK BAT DKO. Thus, PERK is dispensable for the resistance to DIO and for the associated systemic metabolic improvements observed in OPA1 BKO mice.

## GDF15 expression in thermogenic adipocytes is not required to regulate energy metabolism, glucose homeostasis, and core body temperature under baseline conditions

Our transcriptomics data show that *Gdf15* mRNA levels are highly induced in OPA1 BKO mice, and its induction is attenuated in the absence of ATF4, suggesting partial dependence on ATF4 for GDF15 regulation (*Figure 2D and E*). Given GDF15's known effects on energy metabolism in rodents (*Tsai*

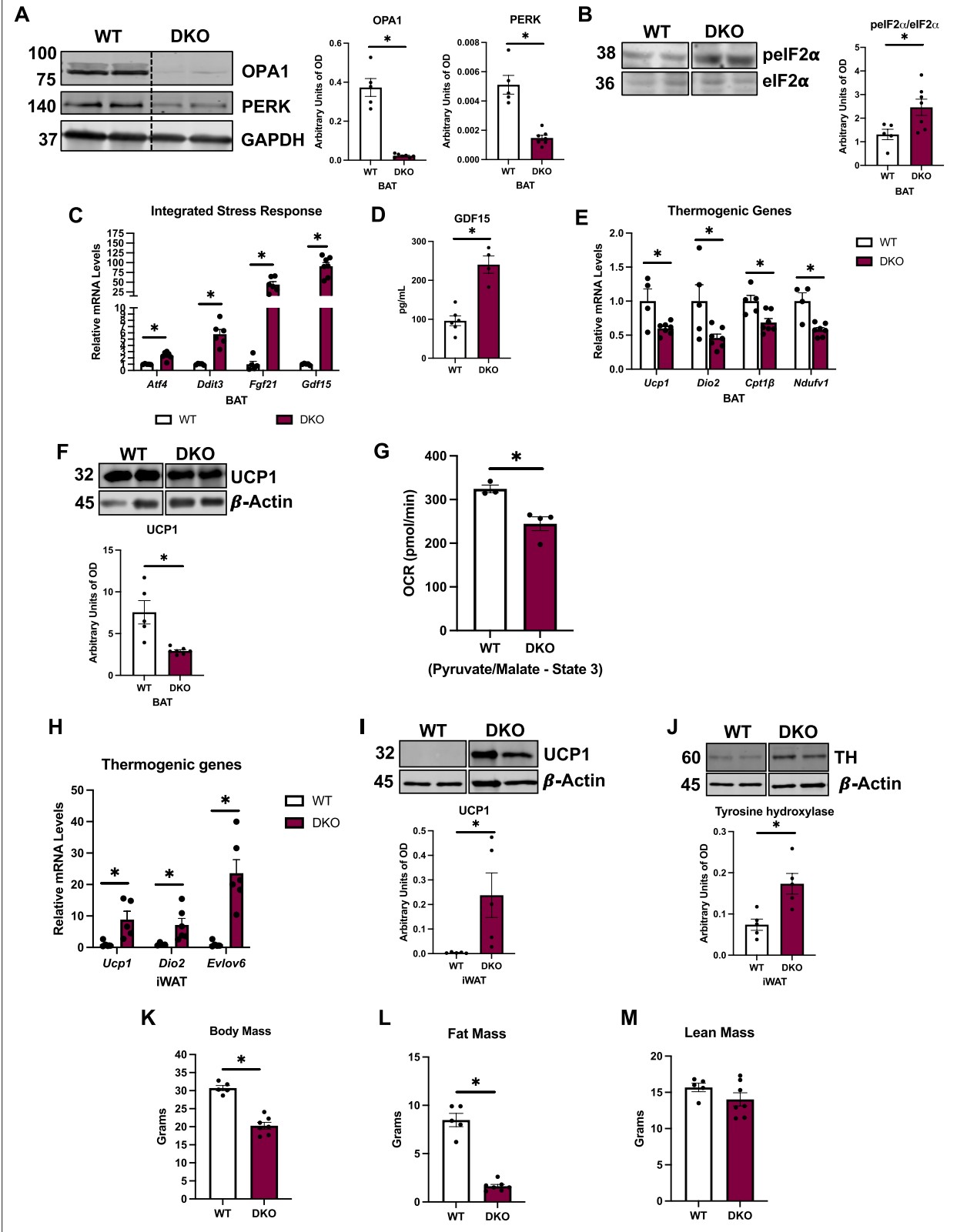

**Figure 3.** PERK is dispensable for the integrated stress response (ISR) activation in OPA1 BKO mice. (**A–G**) Data collected from 8-week-old OPA1/PERK BAT double-knockout (DKO) male and female mice. (**A**) Representative immunoblots for OPA1 and PERK protein in brown adipose tissue (BAT) normalized to GAPDH and their respective densitometric quantification. (**B**) Representative immunoblots for phospho-eIF2α (peIF2α) normalized to total eIF2α protein levels in BAT and the respective densitometric quantification. Optical density (OD). (**C**) Relative mRNA expression of the ISR genes

*Figure 3 continued on next page*

*Figure 3 continued*

*Atf4*, *Ddit3*, *Fgf21,* and *Gdf15* normalized to *Tbp* expression in BAT. (**D**) GDF15 serum levels (ad libitum-fed conditions). (**E**) Relative mRNA expression of the thermogenic genes normalized to tata box protein (*Tbp*) expression in BAT. (**F**) Representative immunoblots for uncoupling protein 1 (UCP1) protein levels in BAT normalized to β-actin and the respective densitometric quantification. (**G**) ADP-stimulated (state 3) pyruvate-malate-supported oxygen consumption rates (OCRs) in mitochondria isolated from BAT. (**H**) Relative mRNA expression of the thermogenic genes normalized to *Tbp* expression in inguinal white adipose tissue (iWAT). (**I**) Representative immunoblots for UCP1 in iWAT normalized to β-actin and the respective densitometric quantification. (**J**) Representative immunoblots for tyrosine hydroxylase (TH) protein levels in iWAT normalized to β-actin and the respective densitometric quantification. OD, optical density. (**K–M**) Data collected in 20-week-old male DKO mice. (**K**) Body mass. (**L**) Total fat mass. (**M**) Total lean mass. Data are expressed as means ± SEM. Significant differences were determined by Student's *t*-test using a significance level of $p < 0.05$. *Significantly different vs. wild-type (WT) mice.

The online version of this article includes the following source data for figure 3:

**Source data 1.** PERK is dispensable for the integrated stress response (ISR) activation in OPA1 BKO mice.

**Source data 2.** Original file with the full raw unedited blot for OPA1 in brown adipose tissue (BAT) of OPA1/PERK double-knockout (DKO) mice.

**Source data 3.** Original file with the full raw unedited blot for PERK in brown adipose tissue (BAT) of OPA1/PERK double-knockout (DKO) mice.

**Source data 4.** Original file with the full raw unedited blot for GAPDH in brown adipose tissue (BAT) of OPA1/PERK double-knockout (DKO) mice.

**Source data 5.** Original file with the full raw unedited blot for peIF2α in brown adipose tissue (BAT) of OPA1/PERK double-knockout (DKO) mice.

**Source data 6.** Original file with the full raw unedited blot for total eIF2α in brown adipose tissue (BAT) of OPA1/PERK double-knockout (DKO) mice.

**Source data 7.** Original file with the full raw unedited blot for total UCP1 in brown adipose tissue (BAT) of OPA1/PERK double-knockout (DKO) mice.

**Source data 8.** Original file with the full raw unedited blot for β-actin for UCP1 in brown adipose tissue (BAT) of OPA1/PERK double-knockout (DKO) mice.

**Source data 9.** Original file with the full raw unedited blot for total UCP1 in inguinal white adipose tissue (iWAT) of OPA1/PERK double-knockout (DKO) mice.

**Source data 10.** Original file with the full raw unedited blot for β-actin for UCP1 in inguinal white adipose tissue (iWAT) of OPA1/PERK double-knockout (DKO) mice.

**Source data 11.** Original file with the full raw unedited blot for total tyrosine hydroxylase (TH) in inguinal white adipose tissue (iWAT) of OPA1/PERK double-knockout (DKO) mice.

**Source data 12.** Original file with the full raw unedited blot for β-actin for tyrosine hydroxylase (TH) in inguinal white adipose tissue (iWAT) of OPA1/ PERK double-knockout (DKO) mice.

*et al., 2018b*), we sought to test the hypothesis that GDF15 induction in OPA1 BKO is required to mediate the resistance to DIO. Prior to analyzing the impact of GDF15 deficiency on the adaptation of BAT to OPA1 loss, we determined whether GDF15 deletion in BAT alone would influence systemic metabolic homeostasis under baseline conditions. Thus, we analyzed mice lacking GDF15 exclusively in UCP1-expressing adipocytes (GDF15 BKO mice). Our data confirms selective downregulation of *Gdf15* mRNA levels in BAT (*Figure 5—figure supplement 1A*), but not in iWAT (*Figure 5—figure supplement 1B*) at room temperature. Under these conditions, deletion of GDF15 in brown adipocytes does not affect circulating levels of GDF15 (*Figure 5—figure supplement 1C*) or core body temperature in mice (*Figure 5—figure supplement 1D*). Furthermore, body mass (*Figure 5—figure supplement 1E*), total fat mass (*Figure 5—figure supplement 1F*), and total lean mass (*Figure 5—figure supplement 1G*) were similar between GDF15 BKO mice and their WT littermate controls. Moreover, glucose homeostasis, as measured by the glucose tolerance test (GTT) (*Figure 5—figure supplement 1H and I*) and fasting blood glucose levels (*Figure 5—figure supplement 1J*), was unaffected by GDF15 deletion in thermogenic adipocytes in young mice (6 wk) fed regular chow. These data indicate that GDF15 deletion in thermogenic adipocytes does not alter systemic metabolism and core body temperature in WT mice fed regular chow under ambient temperature conditions.

## OPA1/GDF15 DKO mice have similar metabolic phenotype as OPA1 BKO mice under baseline conditions

Our data confirmed that increased GDF15 circulating levels in OPA1 KO mice were derived from BAT. mRNA expression demonstrated a significant reduction in *Opa1* and *Gdf15* levels in BAT of OPA1/ GDF15 DKO mice (*Figure 5A*), which completely normalized GDF15 serum levels (*Figure 5B*). In young mice (6–8 wk), BAT mass was slightly elevated in DKO mice (*Figure 5C*). Morphologically, BAT tissue had an increased number of adipocytes with unilocular lipid droplets (*Figure 5D*), indicating

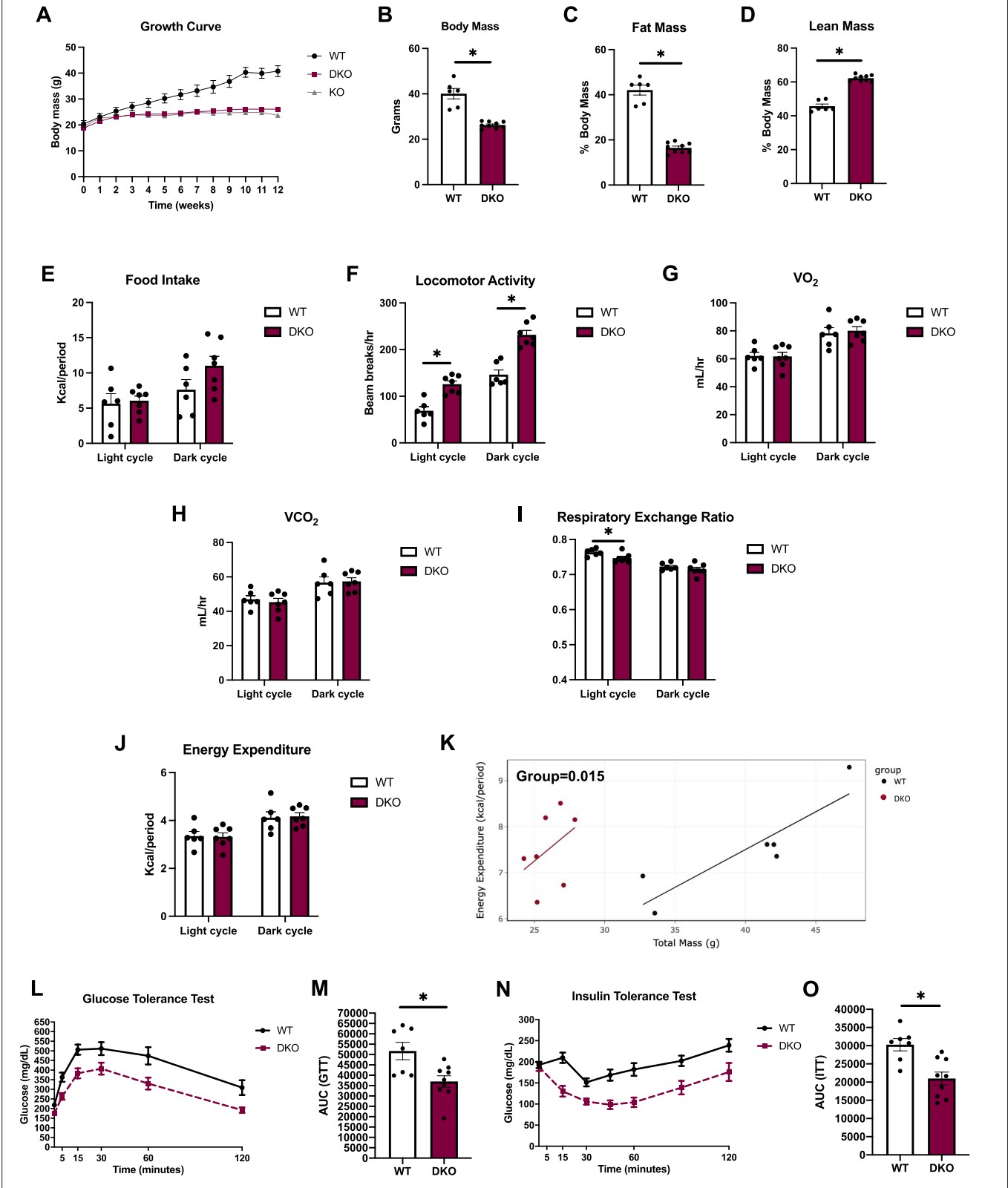

**Figure 4.** OPA1/PERK brown adipose tissue (BAT) double-knockout (DKO) mice are resistant to DIO and insulin resistance. (**A–O**) Data collected in OPA1/PERK BAT DKO male mice fed a high-fat diet (HFD) for 12 wk. (**A**) Body weight curve comparing wild-type (WT), OPA1 BKO, and OPA1/PERK BAT DKO mice. (**B**) Final body weight. (**C**) Percent fat mass. (**D**) Percent lean mass. (**E–K**) Indirect calorimetry data represented as the average for the light and dark cycles during the last 48 hr of data recording. (**E**) Food intake. (**F**) Locomotor activity. (**G**) Oxygen consumption (VO₂). (**H**) Carbon dioxide

*Figure 4 continued on next page*

*Figure 4 continued*

production ($VCO_2$). (**I**) Respiratory exchange ratio. (**J**) Energy expenditure. (**K**) ANCOVA of energy expenditure as a function of body mass. (**L**) Glucose tolerance test (GTT). (**M**) Area under the curve (AUC) for the GTT. (**N**) Insulin tolerance test (ITT). (**O**) AUC for the ITT. Data are expressed as means ± SEM. Significant differences were determined by Student's *t*-test using a significance level of $p < 0.05$. *Significantly different vs. WT mice.

The online version of this article includes the following figure supplement(s) for figure 4:

**Figure supplement 1.** Food consumption locomotor activity and energy expenditure in OPA1/PERK brown adipose tissue (BAT) double-knockout (DKO) mice.

increased whitening, and BAT mitochondria appeared smaller, less electron dense, and had disrupted cristae structure (*Figure 5E*), that were similar to changes observed in OPA1 BKO mice (*Pereira and McFarlane, 2019*). Expression of thermogenic genes at baseline conditions was unchanged in BAT of DKO, except for a small increase in *Ucp1* mRNA levels (*Figure 5F*). However, UCP1 protein levels in BAT were comparable between WT and DKO mice (*Figure 5G*). Similar to OPA1 BKO mice, mitochondrial OCRs were significantly reduced in mitochondria isolated from BAT (*Figure 5H*) and mRNA expression of the ISR genes *Atf4*, *Ddit3*, and *Fgf21* was elevated in DKO mice (*Figure 5I*), which correlated with increased FGF21 serum levels (*Figure 5J*). Compensatory browning of iWAT, as demonstrated by histology (*Figure 5K*), elevated expression of thermogenic genes (*Figure 5L*) and increased UCP1 protein levels (*Figure 5M*) in iWAT, persisted upon GDF15 deletion in the OPA1 BKO background. Tyrosine hydroxylase protein levels, a proxy for sympathetic activation, were elevated in iWAT from DKO mice under baseline conditions (*Figure 5N*), as was observed in OPA1 BKO mice (*Pereira et al., 2021*). Under isocaloric and room temperature conditions, body mass was unchanged at 6 and 10 wk of age but was significantly reduced in 20-week-old DKO mice relative to their WT littermate controls (*Figure 5O*). To test whether GDF15 is required to mediate changes in energy metabolism in OPA BAT KO mice under baseline conditions, 10–12-week-old-mice were housed at thermoneutral conditions (30°C) for 7 d prior to having indirect calorimetry assessed in metabolic chambers. While food intake was unchanged in OPA1 BKO mice (*Figure 1B*), it was slightly reduced in DKO mice during the light cycle (*Figure 5P* and *Figure 5—figure supplement 2A and B*). Locomotor activity (*Figure 5Q* and *Figure 5—figure supplement 2C*) and averaged energy expenditure were similar between genotypes (*Figure 5R* and *Figure 5—figure supplement 2D*). Although body mass was unchanged at this time point, oxygen consumption was slightly increased in DKO mice during the dark cycle (*Figure 5S*), indicating increased metabolic rates. Indeed, ANCOVA shows a significant group effect for the relationship between oxygen consumption and body mass (*Figure 5—figure supplement 2E*), as was also observed in OPA1 BKO mice (*Pereira et al., 2021*). Carbon dioxide production was similar between genotypes (*Figure 5T*), but respiratory exchange ratio, which was increased in OPA1 BKO mice under the same conditions (*Figure 1F*), was reduced in DKO mice during the light cycle (*Figure 5U* and *Figure 5—figure supplement 2F*). These data suggest that GDF15 is largely dispensable in regulating resting metabolic rates in OPA1 BKO mice under baseline conditions, but surprisingly, normalization of GDF15 circulating levels led to reduced food intake and switched fuel preference during the light cycle. Glucose homeostasis as measured by the GTT (*Figure 5—figure supplement 2G and H*) and fasting glucose levels (*Figure 5—figure supplement 2I*) was similar between DKO mice and their WT controls.

## GDF15 partially mediates the resistance to DIO and is required to improve glucose homeostasis in OPA1 BKO mice

Given the role of GDF15 in energy homeostasis (*Chung et al., 2017b*; *Chrysovergis et al., 2014*) and the increase in GDF15 circulating levels in OPA1 BKO mice, we tested the hypothesis that BAT-derived GDF15 is required to mediate resistance to DIO in OPA1 BKO mice. Male WT and DKO mice were fed a HFD (60% calories from fat) for 12 wk. Body mass gain over time was significantly attenuated in DKO relative to their WT littermate controls, but DKO mice gained significantly more weight than OPA1 BKO mice after 12 wk of high-fat feeding (*Figure 6A*). Final body mass (*Figure 6B*) and total fat mass (*Figure 6C*) were also significantly reduced in DKO mice relative to WT mice, while total lean mass was unchanged (*Figure 6D*). These data suggest that GDF15 partially mediates the resistance to DIO observed in OPA1 BKO mice. To gain insight into the mechanisms contributing to these changes in weight gain, we placed a subset of high-fat-fed WT and DKO in metabolic chambers. In contrast

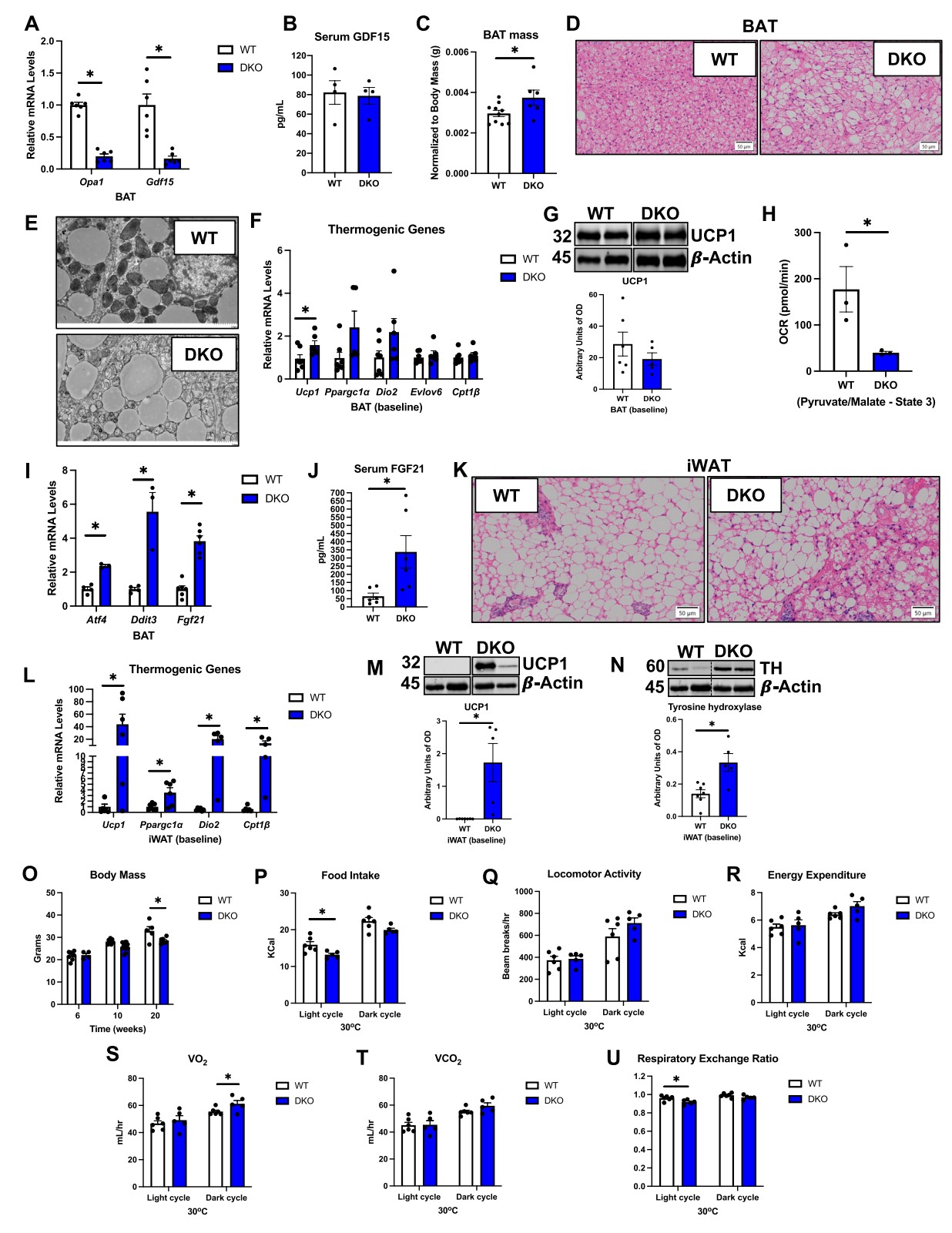

**Figure 5.** OPA1/GDF15 double-knockout (DKO) mice have similar metabolic phenotype as OPA1 BKO mice under baseline conditions. (**A–O**) Data collected in OPA1/GDF15 brown adipose tissue (BAT) DKO mice under baseline conditions (room temperature or 30°C/regular chow). (**A**) Relative mRNA expression of *Opa1* and *Gdf15* in BAT normalized to tata box protein (*Tbp*) expression. (**B**). GDF15 serum levels (ad libitum-fed conditions). (**C**) BAT mass relative to body mass in 8-week-old mice. (**D**) Representative images of H&E-stained histological sections of BAT (n = 3). Scale bar = 50 μm.

*Figure 5 continued on next page*

*Figure 5 continued*

(**E**) Electron micrographs of BAT. Scale bar = 2 μm. (**F**) Relative mRNA expression of the thermogenic genes in BAT normalized to *Tbp* expression. (**G**) Representative immunoblots for UCP1 in BAT normalized to β-actin, and their respective densitometric quantification. Optical density (OD). (**H**) ADP-stimulated (state 3) pyruvate-malate-supported oxygen consumption rates (OCRs) in mitochondria isolated from BAT. (**I**) Relative mRNA expression of the integrated stress response (ISR) genes *Atf4*, *Ddit3*, and *Fgf21* normalized to *Tbp* expression. (**J**) FGF21 serum levels (ad libitum-fed conditions). (**K**) Representative images of H&E-stained histological sections of inguinal white adipose tissue (iWAT) (n = 3). Scale bar = 50 μm. (**L**) Relative mRNA expression of the thermogenic genes in iWAT normalized to *Tbp* expression. (**M**) Representative immunoblots for UCP1 in iWAT normalized to β-actin and their respective densitometric quantification. (**N**) Representative immunoblots for tyrosine hydroxylase (TH) in iWAT normalized to β-actin and their respective densitometric quantification. OD, optical density. (**O**) Body mass at 8, 10, and 20 wk of age. (**P–U**) Indirect calorimetry data represented as the average for the light and dark cycles during the last 48 hr of data recording in male mice around 10–12 wk of age. (**P**) Food intake. (**Q**) Locomotor activity. (**R**) Energy expenditure. (**S**) Oxygen consumption ($VO_2$). (**T**) Carbon dioxide production ($VCO_2$). (**U**) Respiratory exchange ratio. Data are expressed as means ± SEM. Significant differences were determined by Student's *t*-test using a significance level of $p<0.05$. *Significantly different vs. wild-type (WT) mice.

The online version of this article includes the following source data and figure supplement(s) for figure 5:

**Source data 1.** OPA1/GDF15 double-knockout (DKO) mice have similar metabolic phenotype as OPA1 BKO mice under baseline conditions (uncropped blots with the relevant bands labeled).

**Source data 2.** Original file with the full raw unedited blot for UCP1 in BAT of OPA1/GDF15 double-knockout (DKO mice).

**Source data 3.** Original file with the full raw unedited blot for β-actin for UCP1 in BAT of OPA1/GDF15 double-knockout (DKO) mice.

**Source data 4.** Original file with the full raw unedited blot for UCP1 in inguinal white adipose tissue (iWAT) of OPA1/GDF15 double-knockout (DKO) mice.

**Source data 5.** Original file with the full raw unedited blot for β-actin for UCP1 in inguinal white adipose tissue (iWAT) of OPA1/GDF15 double-knockout (DKO) mice.

**Source data 6.** Original file with the full raw unedited blot for tyrosine hydroxylase (TH) in inguinal white adipose tissue (iWAT) of OPA1/GDF15 double-knockout (DKO) mice.

**Source data 7.** Original file with the full raw unedited blot for β-actin for tyrosine hydroxylase (TH) in inguinal white adipose tissue (iWAT) of OPA1/GDF15 double-knockout (DKO) mice.

**Figure supplement 1.** GDF15 expression in wild-type thermogenic adipocytes does not influence energy metabolism, glucose homeostasis and core body temperature under baseline conditions.

**Figure supplement 2.** Metabolic phenotyping of OPA1/GDF15 double-knockout (DKO) mice under baseline conditions.

to OPA1 BKO mice, in which food intake was increased (*Figure 1C*), food intake in DKO mice tended to be lower (*Figure 6E* and *Figure 6—figure supplement 1A and B*). Also, in contrast to OPA1 BKO, locomotor activity (*Figure 6F* and *Figure 6—figure supplement 1C and D*) was unchanged between WT and DKO mice. Energy expenditure was also similar between genotypes (*Figure 6G* and *Figure 6—figure supplement 1E and F*). While in OPA1 BKO mice ANCOVA revealed a near significant group effect for energy expenditure as a function of body mass (*Figure 1Q*), in the DKO mice, ANCOVA detected no significant differences for the group effect (*Figure 6—figure supplement 1G*). The averaged oxygen consumption (*Figure 5H*) and $CO_2$ production (*Figure 5I*) for the light and dark cycles were similar between genotypes. Interestingly, the increase in RER observed in OPA1 BKO mice (*Figure 1O*) was no longer observed in the absence of BAT GDF15 (*Figure 6J* and *Figure 6—figure supplement 1H*). These data suggest that GDF15 is required to mediate the increase in resting metabolic rates observed in OPA1 BKO mice and regulates fuel preference. Although body mass was significantly reduced in DKO mice relative to WT mice, glucose homeostasis, as demonstrated by the GTT (*Figure 6K and L*) and fasting glucose levels (*Figure 6M*), was similarly impaired in WT and DKO mice. Likewise, diet-induced hepatic triglyceride accumulation was comparable between WT and DKO mice (*Figure 6N*). However, insulin sensitivity was significantly improved in DKO mice, as shown by the decreased serum fasting insulin levels (*Figure 6O*) and reduced area under the curve for the insulin tolerance test (ITT) (*Figure 6P and Q*). Although under baseline conditions the expression of thermogenic genes in BAT was largely preserved, upon high-fat feeding, several thermogenic genes were downregulated (*Figure 6R*) and UCP1 protein levels were reduced in BAT (*Figure 6*) of DKO mice. In iWAT, *Ucp1* mRNA levels were elevated in DKO mice (*Figure 6T*), but UCP1 protein levels, which were highly induced under baseline conditions, were undetectable after high-fat feeding in DKO mice (*Figure 6U*). Nonetheless, UCP1 protein levels were significantly induced in iWAT of OPA1 BKO mice after 12 wk on HFD (*Figure 6—figure supplement 1I*). Furthermore, expression of the calcium pump Serca1, which is involved in non-shivering thermogenesis and was recently shown

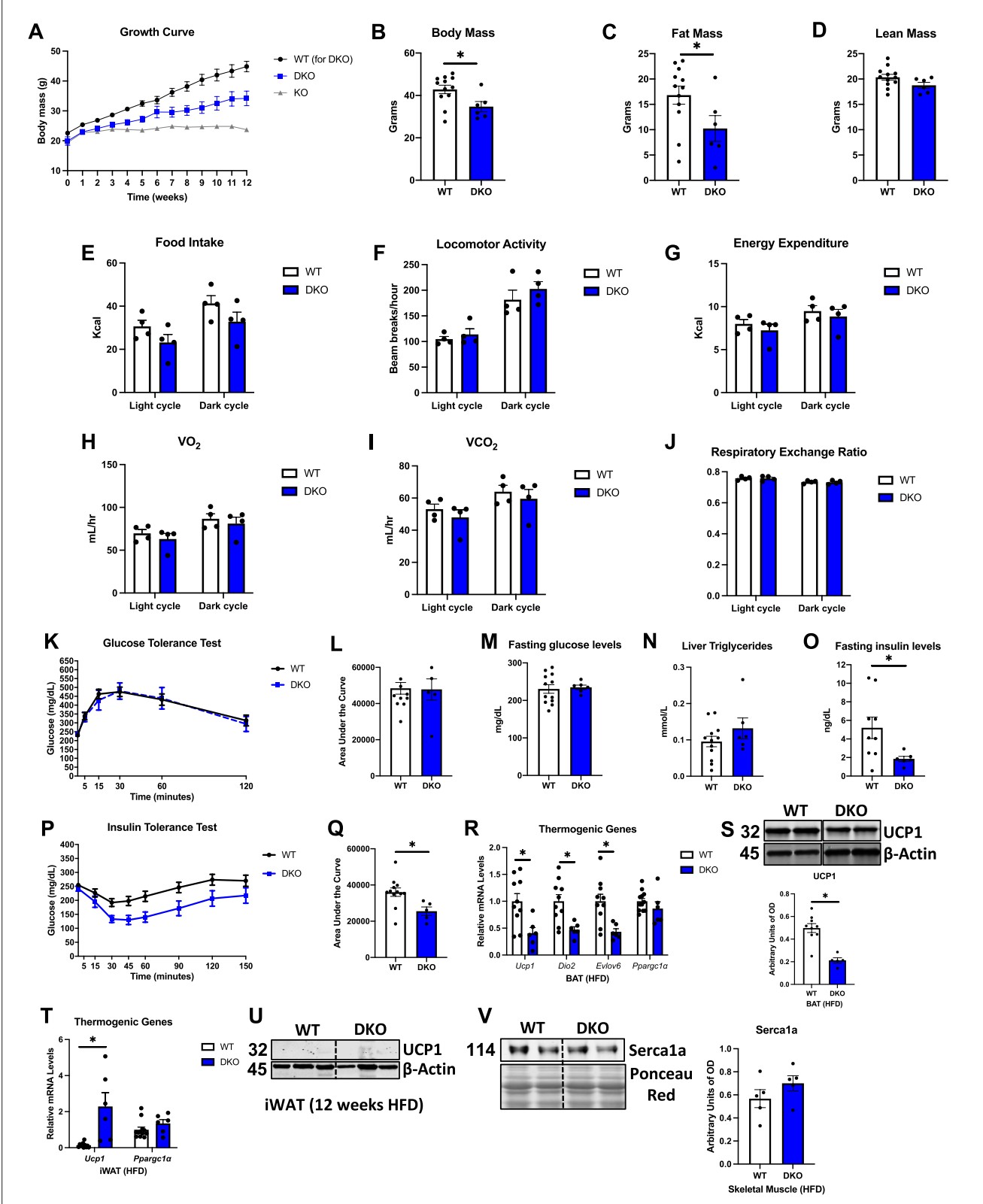

**Figure 6.** GDF15 partially mediates the resistance to diet-induced obesity (DIO) and is required to improve glucose homeostasis in OPA1 BKO mice. (**A–O**) Data collected in OPA1/GDF15 brown adipose tissue (BAT) double-knockout (DKO) male mice fed a high-fat diet (HFD) for 12 wk. (**A**) Body weight curve comparing wild-type (WT), OPA1 BKO, and OPA1/GDF15 BAT DKO mice. (**B**) Final body mass. (**C**) Total fat mass. (**D**) Total lean mass. (**E–J**) Indirect calorimetry data represented as the average for the light and dark cycles during the last 48 hr of data recording. (**E**) Food intake. (**F**) Locomotor activity.

*Figure 6 continued on next page*

*Figure 6 continued*

(**G**) Energy expenditure. (**H**) Oxygen consumption (VO₂). (**I**) Carbon dioxide production (VCO₂). (**J**) Respiratory exchange ratio. (**K**) Glucose tolerance test (GTT). (**L**) Area under the curve (AUC) for the GTT. (**M**) Fasting glucose levels. (**N**) Liver triglycerides levels. (**O**) Fasting insulin levels. (**P**) Insulin tolerance test (ITT). (**Q**) AUC for the ITT. (**R**) Relative mRNA expression of thermogenic genes in BAT normalized to tata box protein (*Tbp*) expression. (**S**) Representative immunoblots for UCP1 in BAT normalized to β-actin and their respective densitometric quantification. Optical Density (OD). (**T**) Relative mRNA expression of thermogenic genes in inguinal white adipose tissue (iWAT) normalized to *Tbp* expression. (**U**) Representative immunoblots for UCP1 in iWAT normalized to β-actin. OD, optical density. (**V**) Representative immunoblots for Serca1a in gastrocnemius muscle normalized to Ponceau red staining and their respective densitometric quantification. OD, optical density. Data are expressed as means ± SEM. Significant differences were determined by Student's *t*-test using a significance level of p<0.05. *Significantly different vs. WT mice.

The online version of this article includes the following source data and figure supplement(s) for figure 6:

**Source data 1.** GDF15 partially mediates the resistance to diet-induced obesity (DIO) and is required to improve glucose homeostasis in OPA1 BKO mice (uncropped blots with the relevant bands labeled).

**Source data 2.** Original file with the full raw unedited blot for UCP1 in brown adipose tissue (BAT) of OPA1/GDF15 double-knockout (DKO) mice after 12 wk on a high-fat diet.

**Source data 3.** Original file with the full raw unedited blot for β-actin for UCP1 in brown adipose tissue (BAT) of OPA1/GDF15 double-knockout (DKO) mice after 12 wk on a high-fat diet.

**Source data 4.** Original file with the full raw unedited blot for UCP1 in inguinal white adipose tissue (iWAT) of OPA1/GDF15 double-knockout (DKO) mice after 12 wk on a high-fat diet.

**Source data 5.** Original file with the full raw unedited blot for β-actin for UCP1 in inguinal white adipose tissue (iWAT) of OPA1/GDF15 double-knockout (DKO) mice after 12 wk on a high-fat diet.

**Source data 6.** Original file with the full raw unedited blot for Serca1 in gastrocnemius muscle of OPA1/GDF15 double-knockout (DKO) mice after 12 wk on a high-fat diet.

**Source data 7.** Original file with the full raw unedited blot for Ponceau red staining for Serca1a in gastrocnemius muscle of OPA1/GDF15 double-knockout (DKO) mice after 12 wk on a high-fat diet.

**Figure supplement 1.** Indirect calorimetry data and immunoblot for thermogenic markers in white adipose tissue (iWAT) and skeletal muscle.

**Figure supplement 1—source data 1.** Immunoblots for thermogenic markers in iWAT and gastrocnemius muscle.

**Figure supplement 1—source data 2.** Original immunoblot image for UCP1 in inguinal white adipose tissue (iWAT) of wild-type (WT) and OPA1 brown adipose tissue (BAT) knockout (KO) mice.

**Figure supplement 1—source data 3.** Original immunoblot image for β-actin for UCP1 in inguinal white adipose tissue (iWAT) of wild-type (WT) and OPA1 brown adipose tissue (BAT) knockout (KO) mice.

**Figure supplement 1—source data 4.** Original immunoblot image for Serca1a in gastrocnemius of wild-type (WT) and OPA1 brown adipose tissue (BAT) knockout (KO) mice.

**Figure supplement 1—source data 5.** Original immunoblot image for Ponceau red staining for Serca1a in gastrocnemius of wild-type (WT) and OPA1 brown adipose tissue (BAT) knockout (KO) mice.

---

to be induced in muscle by GDF15 administration (*Wang et al., 2023*), was induced in gastrocnemius muscle of OPA1 BKO mice (*Figure 6—figure supplement 1J*), but not in DKO mice (*Figure 6V*). Our data suggest that BAT-derived GDF15 is required to sustain browning of iWAT under obesogenic conditions in the OPA1 BKO background and likely regulates calcium cycling in muscle contributing to the increased metabolic rates in these mice.

## GDF15 is required to regulate core body temperature in cold-exposed OPA1 BKO mice

Expression of thermogenic genes is reduced in BAT of DKO mice after DIO (*Figure 6R*), suggesting that thermogenic activation of BAT is impaired in these animals. We, therefore, tested whether core body temperature would be affected in DKO mice at thermoneutrality and in response to cold stress. After 7 d at 30°C, core body temperature, as measured by telemetry, was similar between WT and DKO mice (*Figure 7A*). Conversely, when housed at 4°C, DKO mice became severely hypothermic, with only two mice surviving past the first 24 hr of cold exposure. Indeed, the averaged core body temperature for the initial 24 hr at 4°C was significantly lower in DKO mice relative to their WT counterparts (*Figure 7B*). To capture the changes in core body temperature in all mice, we also plotted the last temperature recorded per mouse, which was greatly reduced in DKO mice (*Figure 7C*). Indirect calorimetry during the first 24 hr of cold exposure showed no significant differences in oxygen consumption for OPA1 BKO (*Figure 7—figure supplement 1A*) or OPA1/GDF15 BAT DKO mice

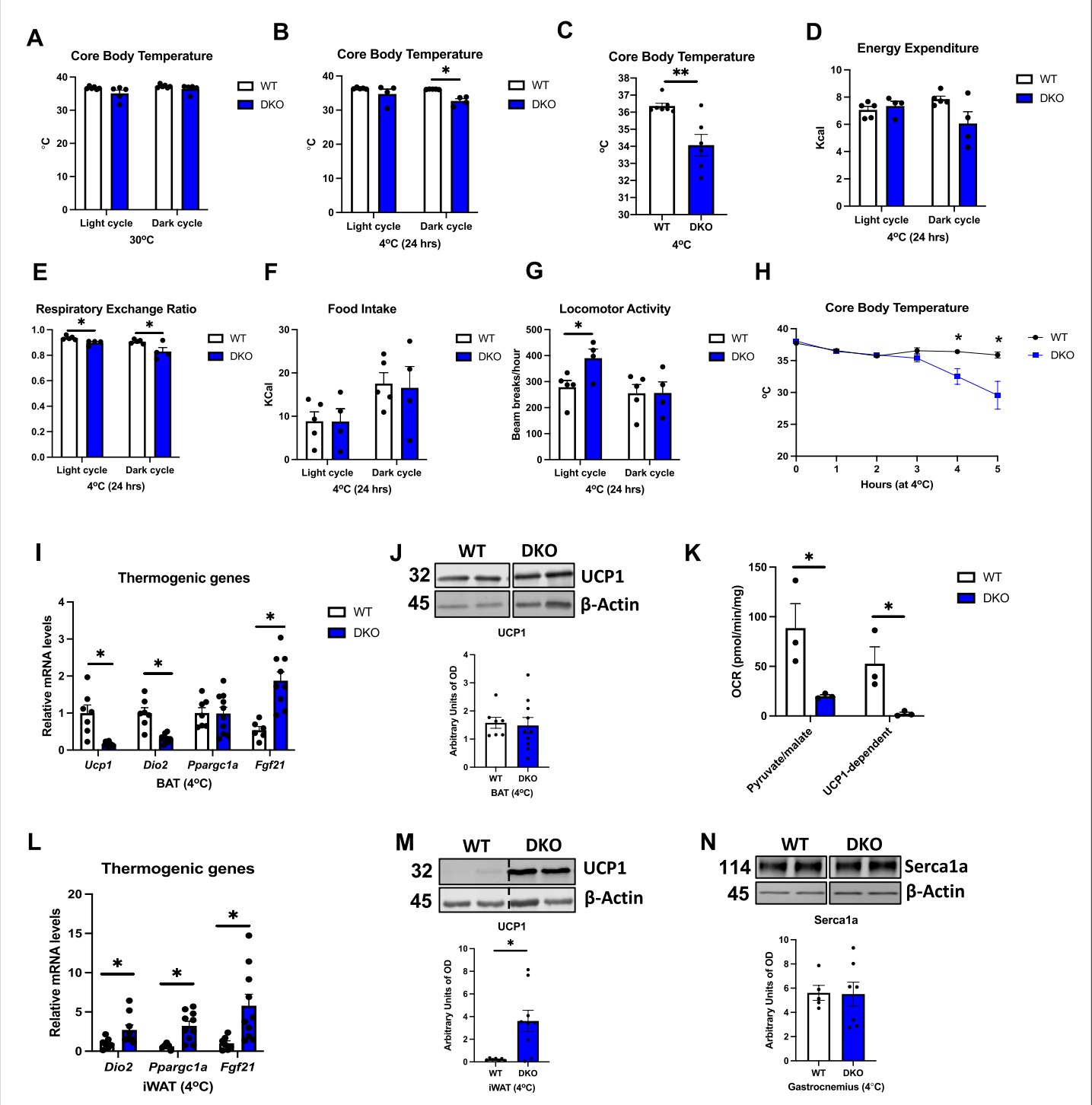

**Figure 7.** GDF15 is required to regulate core body temperature in cold-exposed OPA1 BKO mice. (**A**) Averaged core body temperature (light and dark cycles) collected from 12-week-old wild-type (WT) and OPA1/GDF15 brown adipose tissue (BAT) double-knockout (DKO) mice after 7 d at 30°C. (**B**) Averaged core body temperature (light and dark cycles) in mice cold exposed for 24 hr (4°C). (**C**) Final core body temperature recorded by telemetry in mice exposed to 4°C in the CLAMS system. (**D–G**) Indirect calorimetry data represented as the average for the light and dark cycles during the first 24 hr of data recording (4°C). (**D**) Energy expenditure. (**E**) Respiratory exchange ratio. (**F**) Food intake. (**G**) Locomotor activity. (**H**) Hourly core body temperatures collected from 12-week-old WT and DKO mice during cold exposure (4°C). (**I–N**) Data collected after 5 hr of cold exposure. (**I**) Relative mRNA expression of thermogenic genes in BAT after 5 hr of cold exposure normalized to tata box protein (*Tbp*). (**J**) Representative immunoblots for UCP1 in BAT normalized to β-actin and their respective densitometric quantification. (**K**) Pyruvate-malate-supported oxygen consumption rates (OCRs) and UCP1-dependent respirations in mitochondria isolated from BAT (baseline conditions). (**L**) Relative mRNA expression of thermogenic genes in

*Figure 7 continued on next page*

*Figure 7 continued*

inguinal white adipose tissue (iWAT) normalized to *Tbp* expression. (**M**) Representative immunoblots for UCP1 in iWAT normalized to β-actin and their respective densitometric quantification. (**N**) Representative immunoblots for Serca1a in gastrocnemius muscle normalized to Ponceau red staining and their respective densitometric quantification. Optical density (OD). Data are expressed as means ± SEM. Significant differences were determined by Student's *t*-test using a significance level of p<0.05. * p <0.05; ** p <0.01. Significantly different vs. WT mice.

The online version of this article includes the following source data and figure supplement(s) for figure 7:

**Source data 1.** GDF15 is required to regulate core body temperature in cold-exposed OPA1 BKO mice (uncropped blots with the relevant bands labeled).

**Source data 2.** Original file with the full raw unedited blot for UCP1 in brown adipose tissue (BAT) of OPA1/GDF15 double-knockout (DKO) mice after cold exposure.

**Source data 3.** Original file with the full raw unedited blot for β-actin for UCP1 in brown adipose tissue (BAT) of OPA1/GDF15 double-knockout (DKO) mice after cold exposure.

**Source data 4.** Original file with the full raw unedited blot for UCP1 in inguinal white adipose tissue (iWAT) of OPA1/GDF15 double-knockout (DKO) mice after cold exposure.

**Source data 5.** Original file with the full raw unedited blot for β-actin for UCP1 in inguinal white adipose tissue (iWAT) of OPA1/GDF15 double-knockout (DKO) mice after cold exposure.

**Source data 6.** Original file with the full raw unedited blot for Serca1 in gastrocnemius muscle of OPA1/GDF15 double-knockout (DKO) mice after cold exposure.

**Source data 7.** Original file with the full raw unedited blot for β-actin for Serca1a in gastrocnemius muscle of OPA1/GDF15 double-knockout (DKO) mice after cold exposure.

**Figure supplement 1.** Indirect calorimetry, food intake, locomotor activity, and skeletal muscle characterization of cold-exposed OPA1/GDF15 brown adipose tissue (BAT) double-knockout (DKO) and OPA1 BAT knockout (KO) mice.

**Figure supplement 1—source data 1.** I Full immunoblot images for tyrosine hydroxylase (TH) and β-actin in inguinal white adipose tissue (iWAT) of wild-type (WT) and OPA1/GDF15 brown adipose tissue (BAT) double-knockout (DKO) mice.

**Figure supplement 1—source data 2.** Original immunoblot image for tyrosine hydroxylase in inguinal white adipose tissue (iWAT) of wild-type (WT) and OPA1/GDF15 brown adipose tissue (BAT) double-knockout (DKO) mice.

**Figure supplement 1—source data 3.** Original immunoblot image for β-actin for tyrosine hydroxylase in inguinal white adipose tissue (iWAT) of wild-type (WT) and OPA1/GDF15 brown adipose tissue (BAT) double-knockout (DKO) mice.

**Figure supplement 1—source data 4.** Original immunoblot image for GLUT1 in gastrocnemius muscle of wild-type (WT) and OPA1/GDF15 brown adipose tissue (BAT) double-knockout (DKO) mice.

**Figure supplement 1—source data 5.** Original immunoblot image for β-actin for GLUT1 in gastrocnemius muscle of wild-type (WT) and OPA1/GDF15 brown adipose tissue (BAT) double-knockout (DKO) mice.

(*Figure 7—figure supplement 1B*), although VO$_2$ started declining in DKO mice towards the end of the dark cycle. The averaged energy expenditure for the light and dark cycles during the first 24 hr of cold exposures were not significantly different between WT and DKO mice (*Figure 7D*). In contrast, respiratory exchange ratio, which was unchanged in OPA1 BKO mice (*Figure 7—figure supplement 1C*), was reduced in DKO mice, demonstrating an increased relative preference for fatty acids as energy substrate (*Figure 7E* and *Figure 7—figure supplement 1D*). Food intake was unchanged in OPA1 BKO (*Figure 7—figure supplement 1E*) and DKO mice (*Figure 7F* and *Figure 7—figure supplement 1F*) relative to their WT controls, while locomotor activity was increased during the light cycle in DKO mice (*Figure 7G* and *Figure 7—figure supplement 1H*), but not in OPA1 BKO mice (*Figure 7—figure supplement 1G*) (mice that died within the first 24 hr of cold exposure were removed from these analyses). In a separate cohort of mice housed at 30°C for 7 d, we used a rectal probe to measure core body temperatures prior to cold exposure (time 0) and hourly after ambient temperature was switched to 4°C (for up to 5 hr). Our data shows a precipitous decline in core body temperature in DKO mice, starting at 4 hr post-cold exposure (*Figure 7H*). This phenotype is the opposite of what we previously reported for OPA1 BKO mice, which had increased core body temperature at thermoneutrality and were better able to maintain core body temperature after cold exposure (*Pereira et al., 2021*). After 5 hr, animals were euthanized, and tissues were harvested for analysis. mRNA levels of *Ucp1* and *Dio2* were significantly reduced in BAT of DKO mice, while *Fgf21* mRNA levels were elevated (*Figure 7I*). Nonetheless, UCP1 protein levels (*Figure 7J*) in BAT were unchanged between genotypes. Although UCP1 levels were also unchanged under baseline conditions (*Figure 5G*), UCP1-dependent respirations were markedly inhibited in DKO mice (*Figure 7K*),

indicating impaired UCP1-mediated thermogenesis. Surprisingly, mRNA expression of thermogenic genes (*Figure 7L*) and protein levels of UCP1 were induced in the iWAT of DKO mice (*Figure 7M*), suggesting increased browning. Because induction of browning of iWAT normally occurs after longer periods of cold exposure (*Fisher et al., 2012*), we believe that this increase in thermogenic markers reflects the high baseline compensatory browning of iWAT already observed in DKO mice housed at room temperature conditions (*Figure 5L–M*). TH levels in iWAT were similar between genotypes, likely because of cold-induced TH levels in WT mice (*Figure 7—figure supplement 1I*). It was recently shown that recombinant GDF15 administration in mice can increase energy expenditure in muscle via induction of calcium cycling pathways (*Wang et al., 2023*). Indeed, we observed that OPA1 BKO mice have higher levels of Serca1a, a marker of muscle thermogenesis, in gastrocnemius muscle (*Figure 7—figure supplement 1J*). However, in DKO mice, protein levels of Serca1a were similar to WT levels in gastrocnemius muscle after acute cold exposure (*Figure 7N*), suggesting that GDF15 is required for Serca1a induction in the OPA1 BKO background. GLUT1 levels, which are also normally induced during muscle-mediated thermogenesis, were comparable between WT and DKO mice after cold exposure (*Figure 7—figure supplement 1J*). Taken together, these data indicate that GDF15 induction in BAT is required to regulate core body temperature in cold-exposed OPA1 BKO mice. Although browning of iWAT was preserved after acute cold exposure, the induction of Serca1a in muscle was prevented in DKO mice. Further studies will be required to determine the mechanisms for GDF15-mediated thermoregulation in OPA1 BKO mice, but our data suggest that muscle thermogenesis via futile calcium cycling might contribute to this phenotype.

## Discussion

In our recently published study, we showed that deletion of the mitochondrial fusion protein OPA1 in brown adipocytes impaired thermogenic capacity in BAT, while paradoxically improving thermoregulation and promoting increased metabolic fitness in mice. These adaptations were mediated by the main effector of the ISR, ATF4, partially via the induction of FGF21 as a batokine. Nonetheless, the resistance to DIO and improvements in glucose homeostasis observed in OPA1-deficient mice occurred in an ATF4-dependent, but FGF21-independent manner (*Pereira et al., 2021*). Moreover, the molecular mechanisms leading to activation of the ISR in OPA1 BKO mice remained incompletely understood. Therefore, in the present study, we sought to investigate the molecular mechanisms mediating ISR induction and the mechanisms downstream of ATF4 promoting resistance to DIO in mice lacking OPA1 in thermogenic adipocytes.

The ISR can be induced by four different eIF2α kinases that act as early responders to disturbances in cellular homeostasis. Each kinase responds to distinct environmental and physiological stresses, which reflects their unique regulatory mechanisms (*Pakos-Zebrucka et al., 2016*). We and others have shown induction of the ISR downstream of mitochondrial stress, but the mechanisms mediating this induction are incompletely understood (*Pereira et al., 2021*; *Ost et al., 2020*; *Forsström et al., 2019*; *Ost et al., 2016*). RNASeq data in OPA1-deficient BAT indicated an increase in ER stress pathways and activation of the UPR in OPA1 BKO mice. Indeed, of the four different ISR kinases, the ER kinase PERK was the only one significantly enriched in our transcriptome analysis of OPA1-deficient BAT. Therefore, we tested the hypothesis that PERK, an eIF2α kinase shared by the UPR and the ISR, is required for ATF4 induction in OPA1-deficient BAT. Our data in mice demonstrated that, although ER stress is increased in response to OPA1 deletion, PERK is dispensable for eIF2α phosphorylation and ISR induction in OPA1-deficient BAT. Therefore, mice concomitantly lacking OPA1 and PERK in BAT display similar improvements in systemic metabolic health as observed in OPA1 BKO mice, including increased metabolic rates, browning of iWAT, and resistance to DIO and IR. These data suggest that an alternative ISR kinase is likely activated downstream of mitochondrial stress to induce the ISR in BAT when OPA1 is deleted. Indeed, a recent study revealed that mitochondrial dysfunction induces different paths to promote ISR activation, depending on both the nature of the mitochondrial stress and the metabolic state of the cell (*Mick et al., 2020*). Moreover, studies suggest that heme-regulated eIF2α kinase (HRI) is required to activate ATF4 in response to mitochondrial stress in HEK293 cells treated with oligomycin (*Guo et al., 2020*), and in embryonic and adult hearts of a mouse model of mitochondrial cardiomyopathy (*Zhu et al., 2022*). Whether the same mechanisms mediate ATF4 induction upon OPA1 deletion in BAT remain to be determined.

Transcriptome analysis also revealed that GDF15 was induced in OPA1-deficient BAT, which correlated with increased GDF15 serum levels. GDF15 is a member of the transforming growth factor-β superfamily. It acts through a recently identified receptor, an orphan member of the GFRα family called GFRAL, and signals through the Ret coreceptor (*Breit et al., 2021*). Several studies have demonstrated that GDF15 mediates its effects on reducing food intake, body weight, and adiposity largely by its actions on regions of the hindbrain, where GFRAL is expressed (*Tsai et al., 2018a*; *Xiong et al., 2017*; *Tsai et al., 2019*; *Tran et al., 2018*). A recent study also reported that the GDF15-GFRAL pathway is required to maintain energy expenditure during caloric restriction by increasing β-adrenergic signaling to skeletal muscle, thereby inducing fatty acid oxidation and futile calcium cycling (*Wang et al., 2023*). Nonetheless, several lines of evidence indicate that GDF15 may also have peripheral effects. Indeed, systemic overexpression of GDF15 prevents obesity and insulin resistance by modulating metabolic activity and enhancing the expression of thermogenic and lipolytic genes in BAT and WAT (*Chrysovergis et al., 2014*; *Macia et al., 2012*; *Kim et al., 2013b*). Some of these effects were independent of changes in food intake, suggesting that GDF15 might also exert its effects through yet unidentified receptors other than GFRAL (*Aguilar-Recarte et al., 2022*). GDF15 can be secreted by various organs such as liver (*Patel et al., 2022*), kidney (*Baek and Eling, 2019*), and skeletal muscle (*Ost et al., 2020*). Recently, studies demonstrated that GDF15 can also be induced in brown adipocytes in response to thermogenic stimuli (*Flicker et al., 2019*; *Campderrós et al., 2019*) and prolonged high-fat feeding (*Patel et al., 2019*). Like FGF21, GDF15 is induced in response to mitochondrial stress downstream of the ISR (*Choi et al., 2020*). Indeed, our data show that *Gdf15* mRNA levels were dramatically reduced in mice lacking both OPA1 and ATF4 in BAT, suggesting that ATF4 may indirectly regulate GDF15 induction in OPA1 BKO mice, perhaps via its downstream target, CHOP, which can bind to the GDF15 promoter (*Chung et al., 2017a*).

Deletion of GDF15 in the OPA1 BKO background completely normalized GDF15 serum levels, confirming GDF15 secretion from BAT. Under baseline conditions, the metabolic phenotype of OPA1 BKO mice was largely unaffected by GDF15 deletion in BAT. Importantly, FGF21 levels remained elevated in DKO mice and browning of iWAT and increased resting metabolic rates under baseline conditions were preserved. These data reinforce the idea that FGF21 plays a predominant role on the baseline phenotype of OPA1 BKO mice, particularly browning of iWAT; however, FGF21 was dispensable to promote resistance to DIO in these mice (*Pereira et al., 2021*). Conversely, under obesogenic conditions, our data revealed that GDF15 partially mediates the resistance to DIO. These changes occurred independently of changes in food intake as DKO mice consumed similar amounts of food relative to their WT controls. Rather, our data support the idea that BAT-derived GDF15 increases metabolic rates in OPA1 BKO mice fed obesogenic diets, thereby attenuating weight gain. Furthermore, although under isocaloric conditions browning was preserved in DKO mice, GDF15 was required to maintain browning of iWAT in OPA1 BKO mice fed HFD. These data suggest that GDF15-mediated browning of iWAT during obesogenic conditions might contribute to the increased metabolic rates and reduced weight gain observed in OPA1 BKO mice, as also proposed elsewhere (*Teijeiro et al., 2021*; *Kirschner et al., 2022*; *Peng et al., 2021*). Furthermore, while the calcium pump Serca1 was induced in the gastrocnemius muscle of OPA1 BKO mice fed HFD, this effect was abrogated in DKO mice. This data suggests that GDF15 may also regulate futile calcium cycling in muscle to increase energy expenditure in OPA1 BKO fed an obesogenic diet. Of note, although GDF15 effects on body weight were only partial, the improvements in glucose disposal and hepatic steatosis observed in OPA1 BKO mice were completely abolished in the absence of BAT GDF15, highlighting additional mechanisms of GDF15-mediated metabolic protection that are independent of its body weight-lowering effects. Paradoxically, insulin sensitivity remained improved.

In the context of hepatic mitochondrial dysfunction, which is associated with increased hepatic FGF21 and GDF15 expression and increased liver-derived GDF15 in the circulation, GDF15 was shown to regulate changes in body and fat mass and protect against hepatic steatosis in DIO but had no effect on glucose disposal. Instead, FGF21 was required to increase insulin sensitivity, energy expenditure, and UCP1-mediated thermogenesis in iWAT of regular chow-fed mice (*Kang et al., 2021*), which is similar to what we previously reported for OPA1 BKO mice (*Pereira et al., 2021*). FGF21 and GDF15 induction and secretion was also reported in a model of mitochondrial stress in adipose tissue (both BAT and WAT). In this study, under regular chow-fed conditions, neither GDF15 nor FGF21 appeared to regulate whole-body metabolism. Conversely, as in OPA1 BKO mice, long-term induction of GDF15

attenuated progression of obesity by increasing energy expenditure in DIO, while FGF21 was dispensable for this phenotype (*Choi et al., 2020*). Together, our study suggests that FGF21 and GDF15 play fundamentally distinct roles as part of the mitochondrial stress response in regulating energy metabolism and glucose homeostasis in different metabolic states. Our data support a predominant role for GDF15 rather than FGF21 in promoting resistance to DIO in response to mitochondrial stress and highlights a role for GDF15 in promoting iWAT browning during DIO and potentially inducing futile calcium cycling in skeletal muscle. Some discrepancies between our model and similar models in the literature may stem from differences in the nature of the mitochondrial stress, level of GDF15 induction, and the fact that in our studies FGF21 (*Pereira et al., 2021*) and GDF15 were deleted in a tissue-specific manner, rather than globally. Also, additional methodological details such as age and duration of feeding might have contributed to differences in our overall conclusions.

Because OPA1 BKO mice have improved thermoregulation, we tested the role of BAT-derived GDF15 for adaptive thermogenesis. The increase in averaged baseline core body temperature observed in OPA1 BKO (*Pereira et al., 2021*) mice was no longer seen when GDF15 was deleted in BAT. Furthermore, in response to cold, mice lacking both OPA1 and GDF15 (DKO) became severely hypothermic as early as 4 hr into the cold exposure (4°C) protocol. Surprisingly, these mice had increased markers of browning in white adipose tissue (WAT) after 5 hr of cold exposure, despite similar degrees of sympathetic activation. Nonetheless, this increase in browning was not sufficient to maintain thermoregulation in the context of impaired BAT thermogenesis. Our data suggests increased futile calcium cycling in skeletal muscle of OPA1 BKO mice, but the induction in Serca1a levels was lost in DKO mice. Thus, in addition to severely impaired UCP1-mediated thermogenesis in BAT, inhibited compensation in muscle non-shivering thermogenesis pathways in DKO mice may contribute to overall reduced thermogenic capacity in these mice, leading to severe cold-induced hypothermia. Therefore, at least in the context of mitochondrial stress, GDF15 expression in thermogenic adipocytes plays a critical role in maintaining core body temperature in cold-exposed mice, likely by regulating UCP1-mediated and UCP1-independent thermogenesis.

In conclusion, our study has ruled out a role for PERK in inducing the ISR in response to OPA1 deletion in BAT. Furthermore, we also unveiled a role for GDF15 in attenuating DIO and improving glucose clearance and hepatic steatosis in OPA1 BKO mice. To our knowledge, this is the first demonstration that BAT-derived GDF15 may play a role in energy metabolism, glucose homeostasis, and thermoregulation in the context of mitochondrial stress. Our data also show that GDF15 may exert its effects independently of changes in food intake but predominantly by increasing energy expenditure. The specific molecular mechanisms downstream of GDF15 regulating these metabolic adaptations and the potential peripheral signaling mechanisms that mediate these effects remain to be elucidated. Furthermore, future studies exploring the role of BAT-derived GDF15 on energy metabolism and thermogenesis under physiological conditions might reveal novel GDF15-mediated effects in metabolic homeostasis.

## Materials and methods
### Mouse models
Experiments were performed in male and/or female mice on a C57Bl/6J background. OPA1$^{fl/fl}$ mice (*Zhang et al., 2011*), FGF21$^{fl/fl}$ mice (*Potthoff et al., 2009*), and ATF4$^{fl/fl}$ mice (*Ebert et al., 2012*) were generated as previously described. GDF15$^{fl/fl}$ mice were generated by Dr. Randy Seeley in the C657Bl/6J background and contain loxP sites flanking exon 2 of the *Gdf15* gene. Briefly, to generate GDF15$^{fl/fl}$ mice, the University of Michigan Transgenic Core injected Cas9 protein (Sigma-Aldrich, St. Louis, MO), editing templates containing LoxP sites (IDT Ltd., Coralville, IA) and two sgRNAs (Synthego, Redwood City, CA) recognizing two sites upstream and downstream of exon 2 of the *Gdf15* gene (target sequence: uuggauucacacaacccuag and aggaaaagggacauacagag) into the pronucleus of fertilized mouse embryos. Embryos were then implanted into pseudo pregnant dams. Resultant pups were screened for the presence of LoxP sites within the *Gdf15* genomic sequence. Surrounding sequences were then amplified and subjected to DNA sequencing. Positive animals were bred to C57Bl6/J mice, and the resultant pups were rescreened and resequenced prior to propagation. Subsequent animals were screened for *Gdf15* Flox by PCR (5′ forward: agccagagtaggacggatga; 5′ reverse: caattctgcttcaacccccg; 3′ forward: tgagcccttgggaggtagag; 3′ reverse: ggccacaaaccactctacga

). Transgenic mice expressing Cre recombinase under the control of the *Ucp1* promoter (Tg (Ucp1-cre)1Evdr) (*Kong et al., 2014*) and PERK [fl/fl] mice (*Eif2ak3[tm1.2Drc]*/J) (*Zhang et al., 2002*) were acquired from the Jackson Laboratories (#024670 and #023066, respectively). Compound mutants were generated by crossing OPA1[fl/fl] mice harboring the *Ucp1* Cre with FGF21[fl/fl], ATF4[fl/fl], PERK[fl/fl], or GDF15[fl/fl] mice. GDF15 BKO mice were generated by crossing GDF15[fl/fl] mice with mice harboring the *Ucp1* Cre. WT controls for each compound mutant were mice harboring the respective homozygous floxed alleles but lacking the *Ucp1* Cre. Mice were weaned at 3 wk of age and were kept on standard chow (2920X Harlan Teklad, Indianapolis, IN). For DIO studies, 6-week-old male mice were fed an HFD (60% kcal from fat; Research Diets D12492) for 12 wk. After 11 wk of high-fed feeding, a subset of mice was placed in the Promethion System (Sable Systems International, Las Vegas, NV) to measure changes in energy metabolism, food intake, and locomotor activity. For the cold exposure experiments, mice were acclimated to 30°C (thermoneutral temperature for mice) for 7 d prior to being cold-exposed. For the 3-day cold exposure studies when core body temperature was monitored by telemetry, indirect calorimetry was performed using the OxyMax Comprehensive Lab Animal Monitoring System (CLAMS, Columbus Instruments International, Columbus, OH). Unless otherwise noted, animals were housed at 22°C with a 12 hr light, 12 hr dark cycle with free access to water and standard chow or special diets. All mouse experiments presented in this study were conducted in accordance with the animal research guidelines from NIH and were approved by the University of Iowa IACUC. Male mice were used for these studies unless otherwise noted.

## Cold exposure experiments

For the 3-day cold exposure experiments, core body temperature was measured by telemetry (Respironics, G2 E-Mitter, Murrysville, PA), as previously described (*Pereira et al., 2021*). Core body temperature and indirect calorimetry were recorded in mice singly housed in the OxyMax Comprehensive Lab Animal Monitoring System (CLAMS, Columbus Instruments International). For the acute cold exposure experiments, 12-week-old mice were initially individually housed in the rodent environmental cabinets (Power Scientific, Inc, Pipersville, PA) at 30°C for 7 d. The initial temperature ($t_0$) was recorded using a rectal probe (Fisher Scientific, Lenexa, KS) at 7 am on day 8, after which the temperature was switched to 4°C. Once the desired temperature was reached, we recorded rectal temperatures hourly for up to 5 hr of cold exposure.

## Glucose and insulin tolerance tests, nuclear magnetic resonance, and serum analysis

GTT, ITT, and measurements of serum insulin and FGF21 levels were performed as previously described by us (*Pereira et al., 2021*). Serum GDF15 was measured using commercially available kits according to the manufacturer's directions (R&D Systems, Minneapolis, MN). Body composition was determined by nuclear magnetic resonance (NMR) in the Bruker Minispec NF-50 instrument (Bruker, Billerica, MA).

## Analysis of triglyceride levels

Hepatic triglyceride levels were measured in mice fed HFD for 12 wk using the EnzyChrom Triglyceride Assay Kit (BioAssay Systems, Hayward, CA), as previously described (*Pereira et al., 2021*).

## RNA extraction and quantitative RT-PCR

Total RNA was extracted from tissues with TRIzol reagent (Invitrogen) and purified with the RNeasy kit (QIAGEN Inc, Germantown, MD). Quantitative RT-PCR was performed as previously described (*Pereira et al., 2021*). Data were normalized to *Gapdh* or *Tbp* expression, and results are shown as relative mRNA levels. qPCR primers were designed using Primer-Blast or previously published sequences (*Kim et al., 2013a*).

## RNA sequencing

Bulk RNA sequencing was performed in BAT of 7-week-old female and male OPA1 BKO mice by the Iowa Institute of Human Genetics: Genomics Division at the University of Iowa. Sequencing libraries were prepped using the Illumina TruSeq mRNA Stranded kit and sequenced on a HiSeq4000. Two approaches were used for read alignment, mapping, and quantification. First, a workflow using HISAT2 (v2.1.0), featureCounts (v1.6.3), and DESeq2 (v1.22.2) was performed (*Kim et al., 2019*; *Liao*

*et al., 2014; Love et al., 2014*). The second approach used pseudo-alignment and quantification with Kallisto (v0.45.0) and DESeq2 for differential expression analysis (*Bray et al., 2016*). Ingenuity Pathway Analysis (IPA) software from QIAGEN was utilized for identification of potentially modified pathways. Data visualization was performed using pheatmap and ggplot2 packages in R. RNA-seq data have been deposited to the GEO database under the accession number GSE218907.

## Western blot analysis

Immunoblotting analysis was performed in BAT, iWAT, and gastrocnemius muscle, as previously described (*Pereira et al., 2021*). Membranes were incubated with primary antibodies overnight and with secondary antibodies for 1 hr at room temperature. Fluorescence was quantified using the LiCor Odyssey imager.

## Mitochondrial isolation

Mitochondrial fraction was isolated from BAT, as previously described (*Garcia-Cazarin et al., 2011*). Briefly, tissue was excised, rinsed in ice-cold PBS, and maintained in ice-cold isolation buffer (500 mM EDTA, 215 mM D-mannitol, 75 mM sucrose, 0.1% free fatty acid bovine serum albumin (BSA), 20 mM HEPES, pH 7.4 with KOH) until ready for homogenization. Bradford assay was performed to determine the protein concentration.

## Oxygen consumption

Mitochondrial oxygen consumption rates were assessed in 50 µg of mitochondrial protein using the Oroboros $O_2K$ Oxygraph system (Oroboros Instruments, Innsbruck, Austria) in buffer Z containing (mM) 110 K-MES, 35 KCl, 1 EGTA, 5 $K_2HPO_2$, 3 $MgCl_2.6H_2O$, and 5 mg/ml BSA (pH 7.4, 295 mosmol/l). The following substrates and nucleotides were utilized: pyruvate (5 mM) + malate (2 mM), followed by addition of ADP (5 mM). To inhibit UCP1, GDP (1 mM) was added after pyruvate/malate. UCP1-dependent respiration was calculated by subtracting the OCRs after GDP addition from the OCR prior to adding GDP.

## Transmission electron microscopy

Electron micrographs of BAT were prepared as previously described (*Pereira et al., 2017*). Briefly, BAT was trimmed into tiny pieces (1 mm in width) using a new blade to minimize mechanical trauma to the tissue. Tissues were fixed overnight (in 2% formaldehyde and 2.5% glutaraldehyde), rinsed (0.1% cacodylate pH 7.2), and stained with increasing concentrations of osmium (1.5, 4, and 6%), and dehydrated with increasing concentrations of acetone (50, 75, 95, and 100%). Samples were then embedded, cured, sectioned, and poststained with uranyl and lead. Sections were then imaged on a Jeol 1230 Transmission electron microscope.

## Histology

Fragments of BAT and iWAT were embedded in paraffin, portioned into 4-µm-thick sections, and stained with hematoxylin-eosin (Fisher, Pittsburgh, PA). Light microscopy was performed using an Olympus BX63 (Olympus, Shinjuku, Tokyo, Japan).

## Data analysis

Unless otherwise noted, all data are reported as mean ± SEM. To determine statistical differences, Student's *t*-test was performed for comparison of two groups, and two-way ANOVA followed by Tukey's multiple-comparison test was utilized when more than three groups were compared. A probability value of $p \leq 0.05$ was considered significantly different. Statistical calculations were performed using the GraphPad Prism Software (La Jolla, CA). The association between oxygen consumption or energy expenditure and body mass was calculated by ANCOVA using the CalR software (*Mina et al., 2018*). The significance test for the 'group effect' determined whether the two groups of interest were significantly different for the metabolic variable selected.

## Acknowledgements

This work was supported by AHA Scientist Development Grant 15SDG25710438 and NIH DK125405 to ROP; the Diabetes Research Training Program funded by the NIH (T32DK112751-01) to SHB and to JJ; and the NIH 1R25GM116686 to LMGP. Metabolic phenotyping was performed at the Metabolic Phenotyping Core at the Fraternal Order of Eagles Diabetes Research Center. RNASeq was performed at the Genomics Division of The Iowa Institute of Human Genetics. University of Iowa Central Microscopy Core Facility, for EM analysis. We thank Dr. Hiromi Sesaki, Dr. Matthew J Potthoff, and Dr. Christopher Adams for graciously sharing the OPA1$^{fl/fl}$, FGF21$^{fl/fl}$, and the ATF4$^{fl/fl}$ mice, respectively.

## Additional information

### Competing interests

Randy J Seeley: R.J.S. has received research support from Fractyl, Novo Nordisk, AstraZeneca, and Eli Lilly. R.J.S. has served on scientific advisory boards for Novo Nordisk, CinRX, Scohia, Fractyl and Structure Therapeutics. R.J.S. is a stakeholder of Calibrate and Rewind. The other authors declare that no competing interests exist.

### Funding

| Funder | Grant reference number | Author |
|---|---|---|
| National Institutes of Health | T32DK112751 | Jayashree Jena |
| National Institutes of Health | 1R25GM116686 | Luis Miguel García-Peña |
| National Institutes of Health | DK125405 | Renata O Pereira |
| American Heart Association | 15SDG25710438 | Renata O Pereira |

The funders had no role in study design, data collection and interpretation, or the decision to submit the work for publication.

### Author contributions

Jayashree Jena, Luis Miguel García-Peña, Conceptualization, Data curation, Formal analysis, Validation, Investigation, Visualization, Methodology, Writing – original draft, Project administration; Eric T Weatherford, Data curation, Software, Formal analysis, Validation, Investigation, Visualization, Methodology, Writing – original draft; Alex Marti, Data curation, Formal analysis, Validation, Investigation, Methodology; Sarah H Bjorkman, Data curation, Formal analysis, Investigation, Methodology; Kevin Kato, Jivan Koneru, Jason H Chen, Data curation, Investigation, Methodology; Randy J Seeley, Resources, Validation, Writing – review and editing; E Dale Abel, Resources, Visualization, Writing – review and editing; Renata O Pereira, Conceptualization, Resources, Formal analysis, Supervision, Funding acquisition, Validation, Visualization, Writing – original draft, Project administration, Writing – review and editing

### Author ORCIDs

Jayashree Jena (iD) http://orcid.org/0000-0001-9929-662X
Luis Miguel García-Peña (iD) http://orcid.org/0000-0001-8718-6490
Kevin Kato (iD) http://orcid.org/0000-0002-8466-7149
E Dale Abel (iD) https://orcid.org/0000-0001-5290-0738
Renata O Pereira (iD) http://orcid.org/0000-0001-5809-4669

### Ethics

This study was performed in strict accordance with the recommendations in the Guide for the Care and Use of Laboratory Animals of the National Institutes of Health. All of the animals were handled according to approved institutional animal care and use committee (IACUC) protocols (#0032294) of the University of Iowa. Every effort was made to minimize suffering.

Decision letter and Author response
Decision letter https://doi.org/10.7554/eLife.86452.sa1
Author response https://doi.org/10.7554/eLife.86452.sa2

## Additional files

### Supplementary files
• MDAR checklist

### Data availability
RNASeq data (GSE218907).

The following dataset was generated:

| Author(s) | Year | Dataset title | Dataset URL | Database and Identifier |
|---|---|---|---|---|
| Pereira RO, Weatherford ET | 2022 | Transcriptome analysis in brown adipose tissue of mice lacking the mitochondrial protein optic atrophy 1 (OPA1) in thermogenic adipocytes | https://www.ncbi.nlm.nih.gov/geo/query/acc.cgi?acc=GSE218907 | NCBI Gene Expression Omnibus, GSE218907 |

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

# Appendix 1

## Appendix 1—key resources table

| Reagent type (species) or resource | Designation | Source or reference | Identifiers | Additional information |
|---|---|---|---|---|
| Strain, strain background (mouse, C57Bl/6J) | Murine Models | Jackson Laboratories | JAX Stock #025124 | Tag (Ucp1-cre)1Evdr; male and female |
| Strain, strain background (mouse, C57Bl/6J) | Murine Models | Jackson Laboratories | JAX Stock #023066 | *Eif2ak3*[tm1.2Drc]/J; male and female |
| Antibody | Anti-OPA1 (mouse polyclonal) | BD Biosciences | #612606 | WB 1:1000, primary |
| Antibody | Anti-GAPDH (rabbit monoclonal) | Cell Signaling Technology | #2118 | WB 1:1000, primary |
| Antibody | Anti-UCP1 (mouse monoclonal) | Abcam | #Ab10983 | WB 1:1000, primary |
| Antibody | Anti-Tyrosine hydroxylase (rabbit polyclonal) | Cell Signaling Technology | #2792 | WB 1:1000, primary |
| Antibody | Anti-β-actin (rabbit polyclonal) | Sigma | #A2066 | WB 1:1000, primary |
| Antibody | Anti-phosphorylated eIF2α serine 51 (rabbit monoclonal) | Cell Signaling Technology | #3597 | WB 1:1000, primary |
| Antibody | Anti-eIF2α (mouse monoclonal) | Santa Cruz Biotechnology | #SC81261 | WB 1:1000, primary |
| Antibody | Anti-PERK (rabbit monoclonal) | Cell Signaling Technology | #C33E10 | WB 1:1000, primary |
| Antibody | Anti-Serca1a (mouse monoclonal) | Santa Cruz Biotechnology | #SC515162 | WB 1:1000, primary |
| Antibody | Anti-GLUT1 (rabbit polyclonal) | Millipore | #07-1401 | WB 1:1000, primary |
| Antibody | IRDye 800CW anti-mouse | LI-COR | #925-32212 | WB 1:10,000, secondary |
| Antibody | Alexa Fluor anti-rabbit 680 | Invitrogen | #A27042 | WB 1:10,000, secondary |
| Commercial assay or kit | RNeasy kit | QIAGEN Inc | #74104 | |
| Commercial assay or kit | EnzyChrom Triglyceride Assay Kit | BioAssay Systems | #ETGA-200 | |
| Commercial assay or kit | Mouse/Rat Fibroblast Growth Factor 21 ELISA | Biovendor | #RD291108200R | |
| Commercial assay or kit | Quantikine GDF15 ELISA | R&D Systems | #MGD150 | |
| Commercial assay or kit | Ultra-Sensitive Mouse Insulin ELISA Kit | Chrystal Chem | #90080 | |
| Commercial assay or kit | High-Capacity cDNA reverse Transcription Kit | Applied Biosystems | #4368814 | |
| Software, algorithm | GraphPad Prism Software | GraphPad Software, La Jolla, CA | Version 8.0.0 for Windows | |
| Other | Chow, standard | Harlan Teklad | 2920X | As described under 'Mouse models' |
| Other | Chow, 60% HFD | Research Diets | D12492 | As described under 'Mouse models' |

