## [Editor Report]

This article’s findings are timely as the hormone GDF15 is being widely studied as a potential obesity and diabetes target. The results will add to this growing literature.

---

## [Decision Letter]

**Decision letter after peer review:**

Thank you for submitting your work entitled "GDF15 Is Required for Cold-Induced Thermogenesis and Contributes to Improved Systemic Metabolic Health Following Loss of OPA1 in Brown Adipocytes" for consideration by *eLife*. Your article has been reviewed by three peer reviewers, one of whom is a member of our Board of Reviewing Editors, and the evaluation has been overseen by David James (Senior Editor). The reviewers opted to remain anonymous.

Essential revisions:

1) The reviewers noted a lack of inclusion of single knockout controls. If this data is available, it should be included.

2) More details regarding metabolic phenotyping should be provided.

*Reviewer #1 (Recommendations for the authors):*

Jena et al. follows up on a previous study on OPA1 knockout in BAT (BKO) tissue and provides mechanistic details on how certain phenotypes in OPA1 BKO are mediated. This paper tests and rules out PERK as a potential mediator of ATF4 activation in OPA1 BKO by creating a OPA1 and PERK double KO (DKO) in BAT. It also assigns the phenotypes resistance to diet induced obesity, improvements in glucose tolerance and cold induced thermogenesis to be GDF15 dependent.

The manuscript uses OPA1/PERK BAT DKO and OPA1/GDF15 BAT DKO, and compares them to wildtype (WT) without single KO comparisons. This needs to be clarified.

1. Is PERK activated in OPA1 BKO? There is no western blot showing increased PERK phosphorylation or any other means of activation.

2. Is increase in serum GDF15 levels due to OPA1 BKO in Figure 1E is it in the physiological range for the receptor GFRAL to be activated? Is giving additional GDF15 to WT or to the OPA1/GDF15 DKO to bring the concentration up to OPA1 BKO levels going to replicate these aspects of the OPA1 KO phenotype?

3. One of their conclusions states that "Our data shows that GDF15 may exert its effects independently of changes in food intake but predominantly by increasing energy expenditure." The energy expenditure data shows no difference between WT in HFD and loss of GDF15 increases energy expenditure in chow feeding. This needs clarification.

*Reviewer #2 (Recommendations for the authors):*

In this manuscript, the authors followup on prior findings in which they showed that mice with a brown fat specific knockout of OPA1 (OPA1 BKO) demonstrate induction of ATF4 and FGF21 secretion and resistance to dietary obesity. Since FGF21 is dispensable for these metabolic benefits, here the authors investigate what mediates the activation of the integrated stress response in OPA1 BKO mice and downstream mediators of the protection from dietary obesity. By making new brown fat-selective double mutant mice, they suggest that PERK is not required for ATF4 induction in this context and that deletion of GDF15 in brown fat partially reverses the protection from dietary obesity seen with OPA deletion.

Strengths of this manuscript include the generation of new brown fat selective double knockout models (OPA1/PERK DKO and OPA1/GDF15 DKO). Moreover, the authors performed phenotyping on both standard chow and high fat diets. Their data indicate that PERK in brown adipocytes is not required for the induction of ATF4 seen in OPA1 deletion. They further show that OPA1 BKO mice with concomitant deletion of GDF15 show attenuated protection from dietary obesity. These mice also have impaired cold tolerance. Weaknesses of the manuscript include the absence of single knockout controls in the experiments presented. In addition, the metabolic phenotyping is not comprehensive and underlying mechanisms for the phenotypes observed are not deeply explored.

1. It is very difficult to interpret the findings of the experiments on the DKO models without single knockout controls. The reader is forced to make a mental comparison to OPA1 BKO mice in a prior paper from this group, but those mice were presumably not studied contemporaneously. Because of this, it is difficult to have confidence that the data support the conclusions made. For example, in Figure 2, the authors make OPA1/PERK DKO mice, and they argue that the increase in GDF15 seen with OPA1 deletion is unchanged (Figure 2D). However, without single KO controls, how can one be sure of this? In addition, the authors argue that OPA1/GDF15 DKO mice show attenuated protection from dietary obesity (Figure 4). Again, single OPA1 KOs controls would be needed to make this conclusion more solid.

2. The metabolic phenotyping is not complete in this manuscript. Specifically, the authors should: (i) show longitudinal weight curves instead of static bar graphs; (ii) provide a more thorough analysis of browning in Figure 2, including UCP1 Westerns, qPCR of other thermogenic genes, and histology/IHC; (iii) the analysis of energy expenditure should show longitudinal data over time in the light and dark cycles, rather than bar graphs.

3. The manuscript does not contain any mechanistic exploration of the findings here. The authors begin to do this in Figure 5, but what do they think explains the fact that OPA1/GDF15 DKO mice show partial protection from DIO, yet are also more cold-sensitive? Additionally, the authors argue that the increased browning in iWAT in DKO mice is due to increased sympathetic activation, but they do not present any data in support of this claim.

4. In Figure 1, panels C and D should be in the same graph.

5. In Figure 1, GDF15 levels are in the ng/ml range, but in Figure 2, they are in the pg/ml range. This should be clarified.

6. The PERK KO in BAT was not nearly as complete as the OPA1 KO. Why do the authors think this is? And could this explain the lack of an effect on ATF4?

7. On page 10 of the text, lines 4-5 (in reference to Figure 4), shouldn't there be a NOT in this sentence?

*Reviewer #3 (Recommendations for the authors):*

The report by Jena et al. describes the observation that genetic inhibition of both OPA1 and GDF15 using UCP1-cre leads to reductions in circulating GDF15 and reductions in body mass. These effects are associated with increases in EE during the dark but not light cycle, lower fat mass and reduced expression of thermogenic genes in WAT. The findings are interesting but there are many incomplete findings presented which require further clarification.

1. The data around the assessment of energy expenditure is incomplete and inconsistent. For example in Figure 2 the authors present the EE relative to body but no other data from this experiment is provided. In Figure 4F they show EE is elevated in the dark cycle but not the light cycle. They then plot VO2 relative to body mass in an n=4 mice and show no difference between groups. It is unclear where this data came from and if it was the same experiment and if so whether this was collected in the dark or light cycle? In both Figure 2 and Figure 4 VO2, VCO2, and RER should be presented as it is unclear whether a shift to Fatty acid oxidation might be the primary contributor to the increase in the calculated EE. In addition the sample size for all measures is very low making it possible the results may be due to a type 1 error.

2. The authors suggest that the increase in EE may be leading to the reduced body mass of the DKO mice. To validate this hypothesis authors should complete pair feeding experiments as it is currently unclear whether these relatively small changes in EE are critical for the phenotype which is central to the primary conclusions.

3. It is now well recognized that housing mice outside of the thermoneutral zone masks changes in adaptive thermogenesis. Authors should complete experiments at thermoneutrality.

4. Throughout the paper there is paucity of information besides the assessment of a few thermogenic genes about the morphology of the brown and white adipose tissue depots. Authors should provide adipose tissue histology images and ideally electron microscopy images detailing whether there may be defects in mitochondrial function.

5. Related to the above the authors suggest reduced WAT browning is the reason for the hypothermia with cold, however, it seems highly unlikely that given the relatively low thermogenic capacity of this tissue that this is the primary driver of the effect. For example, what about the very large reduction in BAT UCP1 in the DKO mice. Alternatively, did the authors examine whether there might be differences in shivering or skeletal muscle function?

---

## [Author Response]

Essential revisions:1) The reviewers noted a lack of inclusion of single knockout controls. If this data is available, it should be included.

We appreciate this recommendation. Although we have recently published our data on the OPA1 BKO mice (DOI: 10.7554/*eLife*.66519), we now include previously unpublished data from an independent cohort of OPA1 BKO in this revised version of our manuscript. Due to time constraints and limited resources, we were unable to conduct de novo studies in additional OPA1 BKO mice, but we have added historical data on these animals that were not previously published. The new Figure 1, Figure 1 figure supplement 1, Figure 6 —figure supplement 1 and Figure 7 —figure supplement 1 show data on body mass, energy homeostasis, indirect calorimetry and browning of WAT in OPA1 BKO mice under normal chow and HFD conditions.

2) More details regarding metabolic phenotyping should be provided.

We have expanded the data on metabolic phenotyping for the different mouse models and added more details about data collection and analysis. Although our main conclusions were unaffected by the additional data, we hope these data will be helpful, and will bring clarity thereby satisfying the reviewers’ concerns.

Reviewer #1 (Recommendations for the authors):Jena et al. follows up on a previous study on OPA1 knockout in BAT (BKO) tissue and provides mechanistic details on how certain phenotypes in OPA1 BKO are mediated. This paper tests and rules out PERK as a potential mediator of ATF4 activation in OPA1 BKO by creating a OPA1 and PERK double KO (DKO) in BAT. It also assigns the phenotypes resistance to diet induced obesity, improvements in glucose tolerance and cold induced thermogenesis to be GDF15 dependent.The manuscript uses OPA1/PERK BAT DKO and OPA1/GDF15 BAT DKO, and compares them to wildtype (WT) without single KO comparisons. This needs to be clarified.1. Is PERK activated in OPA1 BKO? There is no western blot showing increased PERK phosphorylation or any other means of activation.

This is an important point, and we appreciate the opportunity to clarify this issue. *Perk* was initially selected from our RNASeq data in BAT of OPA1 BAT KO. Of all integrated stress response (ISR) kinases, PERK was the only one significantly enriched in BAT of OPA1 BAT KO mice. This data is now included in the revised manuscript.

2. Is increase in serum GDF15 levels due to OPA1 BKO in Figure 1E is it in the physiological range for the receptor GFRAL to be activated?

Under physiological conditions, such as after prolonged high-fat feeding, GDF15 circulating levels in mice usually double, going from ~ 100pg/mL to ~ 200 pg/mL (DOI: 10.1016/j.cmet.2018.12.016). Thus, our data suggest that the increase observed in OPA1 BKO mice (Figure 2) is within the physiological range. Importantly, administration of recombinant GDF15 within the physiological range does not lead to changes in food intake, whereas at a higher concentration (~5000 pg/mL), food intake is reduced, suggesting GDF15-mediated activation of GFRAL (DOI: 10.1016/j.cmet.2018.12.016). Although not directly tested in this study, we believe that the increase in GDF15 levels observed in OPA1 BKO mice is likely insufficient to activate GFRAL signaling in the hindbrain to reduce food intake, which might explain why we do not see suppression of food intake associated with increased GDF15 levels in our model. Indeed, it has been proposed that the anorectic effect of GDF15 involving GFRAL requires a threshold of circulating GDF15 levels that is ≥400 pg/ml in mice, with lower values not affecting appetite (10.15252/embr.201948804). Furthermore, several lines of evidence indicate that GDF15 may also have peripheral effects. Indeed, GDF15 increases thermogenesis, lipid catabolism, and mitochondrial oxidative phosphorylation independently of changes in food intake (doi.org/10.1083/jcb.201607110), suggesting that GDF15 might also exert its effects through receptors other than GFRAL and independently of appetite reduction (https://doi.org/10.1016/j.tem.2022.08.004). We have expanded our discussion to address this issue.

Is giving additional GDF15 to WT or to the OPA1/GDF15 DKO to bring the concentration up to OPA1 BKO levels going to replicate these aspects of the OPA1 KO phenotype?

These are interesting experiments to pursue. Unfortunately, due to time constraints, we were unable to conduct these studies. It is likely that restoring GDF15 circulating levels to the levels reported in OPA1 KO mice might replicate some of the OPA1 KO phenotype, particularly those shown to be mediated by GDF15. Future studies using mice lacking GFRAL should clarify which aspects of the OPA1 BAT KO phenotype are regulated by GDF15’s central effects, and which ones require GDF15’s action locally in BAT or possibly in other tissues.

Of note, the degree of GDF15 induction in OPA1 KO mice under baseline conditions is similar to that caused by cold stress or high-fat feeding in WT mice, and is not further induced in response to these stressors.

3. One of their conclusions states that "Our data shows that GDF15 may exert its effects independently of changes in food intake but predominantly by increasing energy expenditure." The energy expenditure data shows no difference between WT in HFD and loss of GDF15 increases energy expenditure in chow feeding. This needs clarification.

We appreciate the reviewer bringing this to our attention. ANCOVA analysis revealed increased energy expenditure and oxygen consumption rates as a function to body mass in OPA1 BAT KO mice under isocaloric conditions and when fed obesogenic diet (DOI: 10.7554/*eLife*.66519 and Figure 1). These data led us to conclude that the increase in metabolic rates resulted in the lean phenotype observed in these mice. Although OPA1/GDF15 DKO mice had increased metabolic rates under isocaloric conditions (Figure 5 and Figure 5 —figure supplement 2), which we believe is mediated by FGF21 (DOI: 10.7554/*eLife*.66519), upon high-fat feeding, the group effect for energy expenditure as a function of body mass was no longer statistically significant (Figure 6 and Figure 6 —figure supplement 1). This suggested to us that GDF15 might regulate changes in energy expenditure in OPA1 BKO mice fed obesogenic diet. Therefore, OPA1/GDF15 DKO mice lack this increase in resting metabolic rates, resulting in increased weight gain relative to OPA1 BKO mice when fed HFD (Figure 6A). Interestingly, OPA1 BAT KO mice fed HFD have high UCP1 expression in iWAT, while OPA1/GDF15 BAT DKO mice lack this adaptation (Figure 6 —figure supplement 1). Furthermore, we observed increased Serca1a levels in the skeletal muscle of OPA1 BKO mice, but not in OPA1/GDF15 DKO mice, suggesting GDF15 might also regulate futile calcium cycling in muscle, as recently demonstrated elsewhere (DOI: 10.1038/s41586-023-06249-4). Although not directly tested in this study, it is possible that GDF15-dependent induction of thermogenic activity in iWAT and in muscle might contribute to the increased metabolic rates observed in OPA1 BAT KO mice during obesogenic conditions contributing to leanness.

Reviewer #2 (Recommendations for the authors):In this manuscript, the authors followup on prior findings in which they showed that mice with a brown fat specific knockout of OPA1 (OPA1 BKO) demonstrate induction of ATF4 and FGF21 secretion and resistance to dietary obesity. Since FGF21 is dispensable for these metabolic benefits, here the authors investigate what mediates the activation of the integrated stress response in OPA1 BKO mice and downstream mediators of the protection from dietary obesity. By making new brown fat-selective double mutant mice, they suggest that PERK is not required for ATF4 induction in this context and that deletion of GDF15 in brown fat partially reverses the protection from dietary obesity seen with OPA deletion.Strengths of this manuscript include the generation of new brown fat selective double knockout models (OPA1/PERK DKO and OPA1/GDF15 DKO). Moreover, the authors performed phenotyping on both standard chow and high fat diets. Their data indicate that PERK in brown adipocytes is not required for the induction of ATF4 seen in OPA1 deletion. They further show that OPA1 BKO mice with concomitant deletion of GDF15 show attenuated protection from dietary obesity. These mice also have impaired cold tolerance. Weaknesses of the manuscript include the absence of single knockout controls in the experiments presented. In addition, the metabolic phenotyping is not comprehensive and underlying mechanisms for the phenotypes observed are not deeply explored.1. It is very difficult to interpret the findings of the experiments on the DKO models without single knockout controls. The reader is forced to make a mental comparison to OPA1 BKO mice in a prior paper from this group, but those mice were presumably not studied contemporaneously. Because of this, it is difficult to have confidence that the data support the conclusions made. For example, in Figure 2, the authors make OPA1/PERK DKO mice, and they argue that the increase in GDF15 seen with OPA1 deletion is unchanged (Figure 2D). However, without single KO controls, how can one be sure of this? In addition, the authors argue that OPA1/GDF15 DKO mice show attenuated protection from dietary obesity (Figure 4). Again, single OPA1 KOs controls would be needed to make this conclusion more solid.

We appreciate this recommendation. Although we understand that ideally the studies in the OPA1 BKO and OPA1/GDF15 DKO mice should have been performed in parallel, our data in OPA1 BKO mice was collected several years ago, and most of it was published in 2021 (DOI: 10.7554/*eLife*.66519). The cost and time associated with repeating all the experiments in parallel with the OPA1/GDF15 and OPA1/PERK DKO studies would have been prohibitive. Nonetheless, we have now added additional data on OPA1 BKO mice that were not previously published. Although we were unable to conduct new studies, we have added historical data (new Figure 1) on body mass and energy homeostasis on OPA1 BAT KO under normal chow and HFD conditions. We hope that the addition of these new data will aid the reviewers in evaluating the different phenotypes described for each mouse model. We believe these data strengthen our conclusions that GDF15 regulates metabolic rates and the resistance to DIO, as well as thermoregulation in cold-exposed OPA1 BKO mice.

2. The metabolic phenotyping is not complete in this manuscript. Specifically, the authors should: (i) show longitudinal weight curves instead of static bar graphs; (ii) provide a more thorough analysis of browning in Figure 2, including UCP1 Westerns, qPCR of other thermogenic genes, and histology/IHC; (iii) the analysis of energy expenditure should show longitudinal data over time in the light and dark cycles, rather than bar graphs.

We appreciate the reviewer’s suggestions. We have now expanded the data on metabolic phenotyping for the different mouse models and added more details about data collection and analysis. Specifically, we: 1. included longitudinal weight curves; 2. provided stronger evidence for browning in both OPA1/PERK and OPA1/GDF15 DKO models; and 3. added longitudinal data over time in the light and dark cycles for energy expenditure and other metabolic parameters.

3. The manuscript does not contain any mechanistic exploration of the findings here. The authors begin to do this in Figure 5, but what do they think explains the fact that OPA1/GDF15 DKO mice show partial protection from DIO, yet are also more cold-sensitive? Additionally, the authors argue that the increased browning in iWAT in DKO mice is due to increased sympathetic activation, but they do not present any data in support of this claim.

We appreciate the opportunity to clarify mechanisms that are driving our phenotype.

In this resubmission, we include new data that we believe offers mechanistic insight to our work. For example, in DIO, although BAT thermogenesis was impaired, OPA1 BAT KO had increased browning of iWAT. This compensatory effect was lost in the absence of GDF15. Furthermore, we observed increased Serca1a levels in the skeletal muscle of OPA1 BKO mice, but not in OPA1/GDF15 DKO mice, suggesting GDF15 might also regulate futile calcium cycling in muscle, as recently demonstrated elsewhere (DOI: 10.1038/s41586-023-06249-4). Although not directly tested in this study, it is possible that GDF15-dependent induction of thermogenic activity in iWAT and in muscle might contribute to the increased metabolic rates observed in OPA1 BAT KO mice during obesogenic conditions contributing to leanness.

Therefore, in the absence of GDF15, when browning is blunted, mice gain significantly more weight (Figure 6 and Figure 6 —figure supplement 1). On the other hand, after acute cold exposure, we see preserved browning of iWAT in DKO mice, while core body temperatures were highly reduced (Figure 7). This suggests that the degree of browning was insufficient to generate enough heat for cold-induced thermogenesis, in light of greatly reduced UCP1-dependent thermogenesis in BAT and inhibited compensatory thermogenic activation in muscle (via futile calcium cycling).

4. In Figure 1, panels C and D should be in the same graph.

We have repeated this analysis in OPA1 BKO and OPA1/ATF4 DKO mice side by side. The data confirms our initial conclusion that GDF15 expression in OPA1 BKO mice is at least partially regulated by ATF4 (Figure 2).

5. In Figure 1, GDF15 levels are in the ng/ml range, but in Figure 2, they are in the pg/ml range. This should be clarified.

This was a mistake on our part and has now been corrected.

6. The PERK KO in BAT was not nearly as complete as the OPA1 KO. Why do the authors think this is? And could this explain the lack of an effect on ATF4?

We agree with the reviewers’ observation that PERK levels are not as reduced in DKO mice as OPA1. We believe this reflects PERK expression in non-adipocytes cells in BAT. However, given the greatly reduced PERK expression, we find it highly unlikely that activation of the residual PERK, could account for the increased eIF2a activation and ATF4 induction. Our interpretation is that reduced total PERK corresponds to reduced levels of phosphorylated PERK. In this scenario, if PERK was required for eIF2a phosphorylation and ISR activation in OPA1 BKO mice, OPA1/PERK DKO mice would have had reduced (or normalized) and not increased eIF2a phosphorylation and ATF4 induction.

7. On page 10 of the text, lines 4-5 (in reference to Figure 4), shouldn't there be a NOT in this sentence?

We understand this may sound confusing. We write “Indeed, the group effect from the ANCOVA analysis for oxygen consumption rates in relation to body weight was not significantly different (Figure 4P), suggesting GDF15 may mediate the increase in metabolic rates observed in OPA1 BAT KO mice in DIO”. We believe, that when examined in light of the new data added on OPA1 BKO mice that this interpretation should make more sense. In the DIO studies, OPA1 BKO mice had elevated metabolic rates, which we believe contributed to the lean phenotype in these mice. Conversely, when GDF15 is deleted in the OPA1 BKO background, mice lack this increase in metabolic rates. Therefore, we concluded that GDF15 is required for this adaptation, and, therefore, upon its deletion, mice gain significantly more weight than the OPA1 BKO.

Reviewer #3 (Recommendations for the authors):The report by Jena et al. describes the observation that genetic inhibition of both OPA1 and GDF15 using UCP1-cre leads to reductions in circulating GDF15 and reductions in body mass. These effects are associated with increases in EE during the dark but not light cycle, lower fat mass and reduced expression of thermogenic genes in WAT. The findings are interesting but there are many incomplete findings presented which require further clarification.1. The data around the assessment of energy expenditure is incomplete and inconsistent. For example in Figure 2 the authors present the EE relative to body but no other data from this experiment is provided. In Figure 4F they show EE is elevated in the dark cycle but not the light cycle. They then plot VO2 relative to body mass in an n=4 mice and show no difference between groups. It is unclear where this data came from and if it was the same experiment and if so whether this was collected in the dark or light cycle? In both Figure 2 and Figure 4 VO2, VCO2, and RER should be presented as it is unclear whether a shift to Fatty acid oxidation might be the primary contributor to the increase in the calculated EE. In addition the sample size for all measures is very low making it possible the results may be due to a type 1 error.

We appreciate the reviewer’s comments. Although we were unable to add more mice to these studies, in this revised manuscript we include additional metabolic data on all mouse models, including VO2, VCO2 and RER. Overall, our data reinforces the conclusion that GDF15 is likely mediating changes to resting metabolic rates in OPA1 BKO mice. Increased RER under obesogenic conditions in OPA1BKO mice, were abrogated in DKO mice, suggesting that GDF15 might also regulate fuel preference. We have included this in our discussion.

2. The authors suggest that the increase in EE may be leading to the reduced body mass of the DKO mice. To validate this hypothesis authors should complete pair feeding experiments as it is currently unclear whether these relatively small changes in EE are critical for the phenotype which is central to the primary conclusions.

Unfortunately, due to time constrains and limited number of animals, we were unable to perform pair-feeding experiments. Nonetheless, our data argues against food intake being the main driver in our phenotype. Despite having elevated GDF15 levels, OPA1 BKO mice have increased food intake under obesogenic conditions (Figure 1), which is not detected in OPA1/GDF15 BKO mice. If food intake was driving the changes in body weight, then DKO mice should weigh less than OPA1 BKO mice, but the opposite is observed. Meanwhile, resting metabolic rates are increased in OPA1 BKO mice fed HFD, but are unchanged in DKO mice. Therefore, our data points to a role for higher energy expenditure as the driver of our phenotype rather than food intake.

3. It is now well recognized that housing mice outside of the thermoneutral zone masks changes in adaptive thermogenesis. Authors should complete experiments at thermoneutrality.

This is an important point. We previously performed experiments in OPA1 BKO mice reared at thermoneutrality (DOI: 10.7554/*eLife*.66519). Our data showed that even under thermoneutral conditions, metabolic rates were elevated in these mice, and body mass was reduced. Based on this initial data, we decided to conduct our studies in DKO mice under room temperature conditions. Nonetheless, in this revised manuscript we show metabolic data collected in mice housed under thermoneutral conditions for 7 days (Figure 1 and Figure 5). Our data show that resting metabolic rates were increased in both OPA1 BKO and OPA1/GDF15 DKO under these conditions.

4. Throughout the paper there is paucity of information besides the assessment of a few thermogenic genes about the morphology of the brown and white adipose tissue depots. Authors should provide adipose tissue histology images and ideally electron microscopy images detailing whether there may be defects in mitochondrial function.

We appreciate the reviewer’s suggestion. We have now included morphological data (histology and TEM for OPA1/GDF15 DKO – Figure 5) and performed mitochondrial respirations in BAT (OPA1/PERK DKO – Figure 3 and OPA1/GDF15 DKO mice – Figure 5).

5. Related to the above the authors suggest reduced WAT browning is the reason for the hypothermia with cold, however, it seems highly unlikely that given the relatively low thermogenic capacity of this tissue that this is the primary driver of the effect. For example, what about the very large reduction in BAT UCP1 in the DKO mice. Alternatively, did the authors examine whether there might be differences in shivering or skeletal muscle function?

We appreciate the opportunity to expand on this point. We indeed observe reduced expression of thermogenic genes in DKO mice in DIO and in response to cold exposure. Although UCP1 protein levels were unchanged under baseline conditions, UCP1-dependent respiration was greatly inhibited. Therefore, it is likely that deletion of GDF15 further impairs BAT thermogenesis rendering DKO mice severely cold intolerant. Although we initially thought that impaired thermogenesis resulted from impaired browning of iWAT, now we show data clearly demonstrating that browning is maintained after cold exposure, and tyrosine hydroxylase levels are unchanged, showing comparable sympathetic activation of iWAT.

Regarding shivering thermogenesis, unfortunately, we were unable to measure this directly, but molecular markers of muscle function (Serca1 and GLUT1) normally affected by cold exposure, were unchanged between cold exposed WT and DKO mice. Interestingly, we now show that Serca1a is induced in gastrocnemius muscle of OPA1 BKO mice (Figure 6 —figure supplement 1), which is inhibited when GDF15 is deleted in BAT (Figure 6V). Therefore, we conclude that although it is unlikely that defects in shivering thermogenesis are contributing to our phenotype, it is possible that non-shivering thermogenesis in muscle via futile calcium cycling may be part of the compensatory mechanisms maintaining core body temperature in cold-exposed OPA1 BKO mice. Indeed, a recent study suggest that recombinant GDF15 induces muscle thermogenesis to maintain energy expenditure in mice (DOI: 10.1038/s41586-023-06249-4).